# LEARNING IN TEMPORALLY STRUCTURED ENVIRONMENTS

**Matt Jones,**[1,2] **Tyler R. Scott,**[1] **Mengye Ren,**[1,3] **Gamaleldin ElSayed,**[1]
**Katherine Hermann,**[1] **David Mayo,**[1,4] **Michael C. Mozer**[1]
[1]Brain Team, Google Research    [2]University of Colorado    [3]NYU    [4]MIT
mcj@colorado.edu dmayo2@mit.edu mengye@cs.nyu.edu
{tylersco,gamaleldin,hermannk,mcmozer}@google.com

## ABSTRACT

Natural environments have temporal structure at multiple timescales. This property is reflected in biological learning and memory but typically not in machine learning systems. We advance a multiscale learning method in which each weight in a neural network is decomposed as a sum of subweights with different learning and decay rates. Thus knowledge becomes distributed across different timescales, enabling rapid adaptation to task changes while avoiding catastrophic interference. First, we prove previous models that learn at multiple timescales, but with complex coupling between timescales, are equivalent to multiscale learning via a reparameterization that eliminates this coupling. The same analysis yields a new characterization of momentum learning, as a fast weight with a negative learning rate. Second, we derive a model of Bayesian inference over $1/f$ noise, a common temporal pattern in many online learning domains that involves long-range (power law) autocorrelations. The generative side of the model expresses $1/f$ noise as a sum of diffusion processes at different timescales, and the inferential side tracks these latent processes using a Kalman filter. We then derive a variational approximation to the Bayesian model and show how it is an extension of the multiscale learner. The result is an optimizer that can be used as a drop-in replacement in an arbitrary neural network architecture. Third, we evaluate the ability of these methods to handle nonstationarity by testing them in online prediction tasks characterized by $1/f$ noise in the latent parameters. We find that the Bayesian model significantly outperforms online stochastic gradient descent and two batch heuristics that rely preferentially or exclusively on more recent data. Moreover, the variational approximation performs nearly as well as the full Bayesian model, and with memory requirements that are linear in the size of the network.

## 1 INTRODUCTION

Many online tasks facing both biological and artificial intelligence systems involve changes in data distribution over time. Natural environments exhibit correlations at a wide range of timescales, a pattern variously referred to as self-similarity, power-law correlations, and $1/f$ noise (Keshner, 1982). This is in stark contrast with the iid environments assumed by many machine learning (ML) methods, and with diffusion or random-walk environments that exhibit only short-range correlations. Moreover, biological learning systems are well-tuned to the temporal statistics of natural environments, as seen in phenomena of human cognition including power laws in learning (Anderson, 1982), power-law forgetting (Wixted & Ebbesen, 1997), long-range sequential effects (Wilder et al., 2013), and spacing effects (Anderson & Schooler, 1991; Cepeda et al., 2008). An important goal is to incorporate similar inductive biases into ML systems for online or continual learning.

This paper analyzes a framework for learning in temporally structured environments, **multiscale learning**, which for neural networks (NNs) can be implemented as a new kind of optimizer. A common explanation for self-similar temporal structure in nature is that it arises from a mixture of events at various timescales. Indeed, many generative models of $1/f$ noise involve summing independent stochastic processes with varying time constants (Eliazar & Klafter, 2009). Accordingly, the multiscale optimizer comprises multiple learning processes operating in parallel at different timescales.

In a NN, every weight $w_j$ is replaced by a family of subweights $\omega_{ij}$, each with its own learning rate and decay rate, that sum to determine the weight as a whole. Learning at multiple timescales is a key idea in several theories in neuroscience, including conditioning (Staddon et al., 2002), learning (Benna & Fusi, 2016), memory (Howard & Kahana, 2002; Mozer et al., 2009), and motor control (Kording et al., 2007), and has also been exploited in ML (Hinton & Plaut, 1987; Rusch et al., 2022). The multiscale learner isolates and simplifies this idea, by assuming knowledge at different timescales evolves independently and that credit assignment follows gradient descent.

The first part of this paper (Sections 2 and 3) proves three other models are formally equivalent to instances of the multiscale optimizer: a new variant of fast weights (cf. Ba et al., 2016; Hinton & Plaut, 1987), the model synapse of Benna & Fusi (2016), and momentum learning (Rumelhart et al., 1986; Qian, 1999). The insight behind these proofs is that each of these models can be written in terms of a linear update rule with diagonalizable transition matrix. Thus the eigenvectors of this matrix correspond to states that evolve independently. By writing the state of the model as a mixture of eigenvectors, we effect a coordinate transformation that exactly yields the multiscale optimizer. These results imply that the complicated coupling among timescales assumed by some models can be superfluous. They also provide a new perspective on momentum learning, with implications for how and when it is beneficial and how it interacts with nonstationarity in the task environment.

In Section 4, we provide a normative grounding for multiscale learning in terms of Bayesian inference over $1/f$ noise. Our starting point is a generative model of $1/f$ noise as a sum of diffusion processes at different timescales. Exact Bayesian inference with respect to this generative process is possible using a Kalman filter (KF) that tracks the component processes jointly (Kording et al., 2007). When learning a single environmental parameter $\theta$, such as mean reward for some action in a bandit task, this amounts to modeling $\theta(t) = \sum_{i=1}^{n} z_i(t)$, where each $z_i$ is a diffusion process with a different characteristic timescale $\tau_i$, and doing joint inference over $\boldsymbol{Z} = (z_1, \ldots, z_n)$.

We then generalize this approach to an arbitrary statistical model, $h(\boldsymbol{x}, \boldsymbol{\theta})$, where $\boldsymbol{x}$ is the input and $\boldsymbol{\theta} \in \mathbb{R}^m$ is a parameter vector to be estimated. For instance, $h$ might be a NN with parameters $\boldsymbol{\theta}$. Our Bayesian model places a $1/f$ prior on $\boldsymbol{\theta}$ (as a stochastic process), by assuming $\boldsymbol{\theta}(t) = \sum_{i=1}^{n} \boldsymbol{z}_i(t)$ for diffusion processes $\boldsymbol{z}_i \in \mathbb{R}^m$ with characteristic timescales $\tau_i$. We then do approximate inference over the joint state $\boldsymbol{Z} = (\boldsymbol{z}_1, \ldots, \boldsymbol{z}_n)$, using an extended Kalman filter (EKF) that linearizes $h$ by calculating its Jacobian at each step (Singhal & Wu, 1989; Puskorius & Feldkamp, 2003). Next, we derive a variational approximation to the EKF that constrains the covariance matrix to be diagonal, and show how it extends the multiscale optimizer. Specifically, writing $w_j$ and $\omega_{ij}$ as the current mean estimates of $\theta_j$ and $z_{ij}$ (for weight $j$ and time scale $i$), the variational update to each $\omega_{ij}$ follows that of the multiscale optimizer, with additional machinery for determining decay rates based on $\tau_i$ and adapting learning rates based on the current prior variance $s_{ij}^2(t)$.

In Section 5, we test our methods in online prediction and classification tasks with nonstationary distributions. In online learning, nonstationarity often manifests as poorer generalization performance on future data versus held-out data from within the training interval. Common solutions are to train on a window of fixed length (to exclude "stale" data) or to use stochastic gradient descent (SGD) with fixed learning rate and weight decay, which leads older observations to have less influence (Ditzler et al., 2015). Here, we demonstrate that performance can be significantly improved by retaining all data and using a learning model that accounts for the temporal structure of the environment. We introduce nonstationarity in our simulations by varying the latent data-generating parameters according to $1/f$ noise. Thus an important caveat is the task domains are matched to the Bayesian model. Notwithstanding, we test robustness by using a different set of timescales for task generation versus learning (Section 5.1), a generative process that mismatches the NN architecture (Section 5.2), and a construction of $1/f$ noise that differs from the sum-of-diffusion process the model assumes (Section 5.3). Results show the Bayesian methods (KF and EKF) outperform windowing and online SGD, as well as a novel heuristic of training the network on all past data with gradients weighted by recency. We also find the variational approximation performs nearly as well as the full model (Section 5.1) and scales well to a multilayer NN trained on real data (Section 5.3).

## 2 MULTISCALE OPTIMIZER

Assume a statistical model $\hat{\boldsymbol{y}}(t) = h(\boldsymbol{x}(t), \boldsymbol{w}(t))$ and loss function $\mathcal{L}(\boldsymbol{y}, \hat{\boldsymbol{y}})$, where $\boldsymbol{x}(t)$ is the input on step $t$, $\boldsymbol{w}(t)$ is the parameter estimate, $\hat{\boldsymbol{y}}(t)$ is the model output, and $\boldsymbol{y}(t)$ is the target output. In a

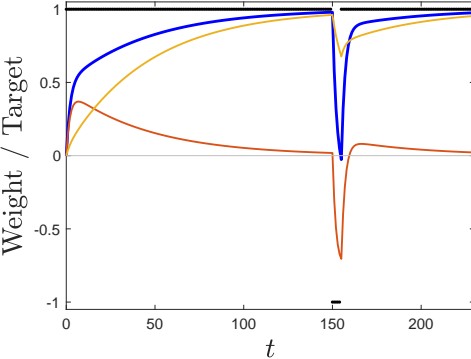

Figure 1: Toy illustration of fast weights. A single weight $w$ (blue) with constant input ($x \equiv 1$) predicts a target signal $T$ (black) with square loss $\mathcal{L} = \frac{1}{2}(T - w)^2$. The weight is a sum of subweights $\omega_{\text{slow}}$ (yellow) and $\omega_{\text{fast}}$ (red). Initial learning is rapid, due to $\omega_{\text{fast}}$. Because of decay and the shared error signal, knowledge is gradually transferred to $\omega_{\text{slow}}$ while $\omega_{\text{fast}}$ returns to zero. When the task switches (trial 151), $\omega_{\text{fast}}$ enables rapid adaptation while long-term knowledge is preserved in $\omega_{\text{slow}}$. Thus the model recovers quickly on the second reversal (compare blue curve beginning on trials 1 vs 156). The general multiscale optimizer extends this idea to an array of faster and slower weights.

NN, $\boldsymbol{w}(t)$ is the vector of current weights. (Under the Bayesian framing in Section 4, $\boldsymbol{w}$ is the mean estimate of the optimal parameters $\boldsymbol{\theta}$.) For exposition, we assume the weights are updated by SGD,

$$\boldsymbol{w}(t+1) = \boldsymbol{w}(t) - \alpha\nabla_{\boldsymbol{w}(t)}\mathcal{L}(\boldsymbol{y}(t), \hat{\boldsymbol{y}}(t)), \tag{1}$$

and we henceforth abbreviate the gradient as $\partial_{\boldsymbol{w}(t)}\mathcal{L}$. However, the following approach can be naturally composed with other optimizers, such as extensions of SGD or Hebbian learning, by replacing $-\alpha\partial_{w(t)}\mathcal{L}$ with the appropriate update term.

The multiscale optimizer is motivated by the assumption that, in online learning tasks, the true or optimal parameters change over time, on multiple timescales. Accordingly, it expands each weight into a sum of subweights, $w_j = \sum \omega_{ij}$, each with a different learning rate $\alpha_i$ and decay rate $\gamma_i$. Here $j$ indexes weights in $\boldsymbol{w}$, and $i$ indexes timescales. The subweights evolve according to:

$$\omega_{ij}(t+1) = \gamma_i\omega_{ij}(t) - \alpha_i\partial_{w_j(t)}\mathcal{L}. \tag{2}$$

Each $\omega_{ij}$ has characteristic timescale $\tau_i := (-\log\gamma_i)^{-1}$. Note that $\partial_{w_j(t)}\mathcal{L} = \partial_{\omega_{ij}(t)}\mathcal{L}$, so one can think of the gradient for $w_j$ being apportioned among the subweights (with total learning rate $\alpha = \sum \alpha_i$), or equivalently of each subweight following its own gradient.

## 2.1 FAST WEIGHTS

A potentially important special case of multiscale learning arises with two timescales, $\boldsymbol{w} = \boldsymbol{\omega}_{\text{slow}} + \boldsymbol{\omega}_{\text{fast}}$. We assume $\gamma_{\text{slow}} = 1$ (no decay) and $\alpha_{\text{fast}} > \alpha_{\text{slow}}$. Thus each $\omega_{\text{slow},j}$ can be thought of as the original weight, which is augmented by $\omega_{\text{fast},j}$, a second channel between the same neurons that both learns and decays rapidly. The fast weight enables the system to adapt quickly to distribution shifts while resisting catastrophic forgetting (Figure 1).

This model is conceptually similar to the fast weights approach of Ba et al. (2016) and Hinton & Plaut (1987). In that work, the weights are updated by a different mechanism (Hebbian learning) than the primary weights, and they act as a memory of recent hidden states in a recurrent network. In the present conception, fast weights optimize the same loss as the primary weights, only with different temporal properties, and they act as a memory for recent learning signals (e.g., loss gradients). Thus they are perhaps better suited for handling distribution shifts of the sort considered here.

## 3 EQUIVALENCE RESULTS

### 3.1 BENNA-FUSI SYNAPSE

Benna & Fusi's (2016) model synapse is designed to capture how biochemical mechanisms in real synapses implement a cascading hierarchy of timescales, and has been adopted in ML for continual reinforcement learning (Kaplanis et al., 2018; 2019). We consider a single weight $w$ in a network, suppressing the index $j$. The Benna-Fusi model assumes that the information in $w$ is maintained in a 1d hierarchy of variables $u_1, \ldots, u_n$, each dynamically coupled to its immediate neighbors:

$$C_1(u_1(t+1) - u_1(t)) = g_1(u_2(t) - u_1(t)) - \partial_{w(t)}\mathcal{L} \tag{3}$$

$$C_k(u_k(t+1) - u_k(t)) = g_{k-1}(u_{k-1}(t) - u_k(t)) + g_k(u_{k+1}(t) - u_k(t)) \tag{4}$$

for $2 \leq k \leq n$, with $g_n = 0$. The external behavior of the synapse comes from $u_1$ alone (i.e., $w = u_1$), while $u_{2:n}$ act as stores with progressively longer timescales.

This update rule can be rewritten as

$$\boldsymbol{u}(t+1) = \boldsymbol{T}\boldsymbol{u}(t) - \boldsymbol{d}(t), \tag{5}$$

with transition matrix $\boldsymbol{T}$ determined by the coefficients in Equations 3 and 4, and external signal $\boldsymbol{d}(t)$ defined by $d_1(t) = \frac{1}{C_1}\partial_{w(t)}\mathcal{L}$ and $d_{2:n} \equiv 0$. It can be shown that the transition matrix is diagonalizable, $\boldsymbol{T} = \boldsymbol{V}\boldsymbol{\Lambda}\boldsymbol{V}^{-1}$, with eigenvalues $\Lambda_{ii} = \lambda_i < 1$ (see Appendix A). We can further enforce $\boldsymbol{V}_{1\cdot} = \boldsymbol{1}$, for a purpose explained below. We refer to the eigenvectors (columns $\boldsymbol{V}_{\cdot i}$) as **modes** of the system, because they are preserved over time up to a scalar. That is, if the initial state is proportional to mode $i$, then in the absence of external signal ($\boldsymbol{d} \equiv 0$), the system will remain in that mode, decaying exponentially with rate factor $\lambda_i$:

$$\boldsymbol{u}(0) \propto \boldsymbol{V}_{\cdot i} \quad \Longrightarrow \quad \forall_t : \boldsymbol{u}(t) = \lambda_i^t \boldsymbol{u}(0) \tag{6}$$

In general, any state can be written uniquely as a linear combination of modes, $\boldsymbol{u} = \sum \omega_i \boldsymbol{V}_{\cdot i} = \boldsymbol{V}\boldsymbol{\omega}$. Therefore, reparameterizing the model as $\boldsymbol{\omega} := \boldsymbol{V}^{-1}\boldsymbol{u}$ yields the simplified update equation:

$$\boldsymbol{\omega}(t+1) = \boldsymbol{\Lambda}\boldsymbol{\omega}(t) + \boldsymbol{V}^{-1}\boldsymbol{d}(t) \tag{7}$$

where $\boldsymbol{V}^{-1}\boldsymbol{d}(t) = \frac{1}{C_1}[\boldsymbol{V}^{-1}]_{\cdot 1}\partial_{w(t)}\mathcal{L}$. Because $\boldsymbol{\Lambda}$ is diagonal, there is no cross-talk between the modes, unlike in the original dynamics. Thus we have derived an instance of the multiscale optimizer, with subweights $\omega_i(t)$, decay rates $\lambda_i$, and learning rates $\frac{1}{C_1}[\boldsymbol{V}^{-1}]_{i1}$. The assumption above, $\boldsymbol{V}_{1\cdot} = \boldsymbol{1}$, implies $w = u_1 = \sum \omega_i$, so the models agree on the external behavior of the weight as a whole. Figure 2 illustrates the translation between the two models.

## 3.2 MOMENTUM LEARNING

The standard rationale for momentum learning is to smooth updates over time, so that oscillations along directions of high curvature cancel out while progress can be made in directions with consistent gradients (Rumelhart et al., 1986). To simplify notation, we again focus on a single weight $w$ in the network, suppressing the index $j$. The momentum $g$ is defined as an exponentially filtered running average of gradients, with weight update determined by current momentum:

$$g(t+1) = \beta g(t) + (1 - \beta)\partial_{w(t)}\mathcal{L} \tag{8}$$

$$w(t+1) = w(t) - \eta g(t+1). \tag{9}$$

This formulation is equivalent to one in which the update $\Delta w(t) = w(t+1) - w(t)$ includes a portion of the previous update: $\Delta w(t) = -\alpha\partial_{w(t)}\mathcal{L} + \beta\Delta w(t-1)$, with $\alpha = \eta(1-\beta)$.

Paralleling the analysis in Section 3.1, we write the state of the momentum optimizer as $[w, g]^\top$ and use Equations 8 and 9 to obtain the update rule:

$$\left[ \begin{array}{c} w(t+1) \\ g(t+1) \end{array} \right] = \left[ \begin{array}{cc} 1 & -\eta\beta \\ 0 & \beta \end{array} \right] \left[ \begin{array}{c} w(t) \\ g(t) \end{array} \right] + \left[ \begin{array}{c} -\eta(1-\beta) \\ (1-\beta) \end{array} \right] \partial_{w(t)}\mathcal{L}. \tag{10}$$

The transition matrix has eigenvectors $[1, 0]^\top$ with eigenvalue 1, and $[1, \frac{1-\beta}{\eta\beta}]^\top$ with eigenvalue $\beta$. Now use this eigenbasis to define a reparameterization:

$$\left[ \begin{array}{c} w \\ g \end{array} \right] = \left[ \begin{array}{cc} 1 & 1 \\ 0 & \frac{1-\beta}{\eta\beta} \end{array} \right] \left[ \begin{array}{c} \omega_{\text{slow}} \\ \omega_{\text{fast}} \end{array} \right]. \tag{11}$$

Substitution into Equation 10 yields the reparameterized update rule:

$$\left[ \begin{array}{c} \omega_{\text{slow}}(t+1) \\ \omega_{\text{fast}}(t+1) \end{array} \right] = \left[ \begin{array}{cc} 1 & 0 \\ 0 & \beta \end{array} \right] \left[ \begin{array}{c} \omega_{\text{slow}}(t) \\ \omega_{\text{fast}}(t) \end{array} \right] - \left[ \begin{array}{c} \eta \\ -\eta\beta \end{array} \right] \partial_{w_t}\mathcal{L}. \tag{12}$$

Thus we recover the fast-weight optimizer, with decay $\gamma_{\text{fast}} = \beta$ and learning rates $\alpha_{\text{slow}} = \eta$ and $\alpha_{\text{fast}} = -\eta\beta$. The negative fast learning rate is perhaps surprising but can be understood as follows: When $\varepsilon_{\text{fast}} < 0$, the subweights learn in opposite directions, with the latent knowledge in $\omega_{\text{slow}}$ overshooting the observable knowledge in $w = \omega_{\text{slow}} + \omega_{\text{fast}}$. As $\omega_{\text{fast}}$ decays toward 0, $w$ catches up to $\omega_{\text{slow}}$, so that the model appears to continue learning from past input, just as it would with momentum. This analysis highlights the contrasting rationales of these two methods: Learning at multiple timescales is motivated by an expectation of positive autocorrelation in the environment, whereas momentum is effective at smoothing out negative autocorrelation in the gradient signal.

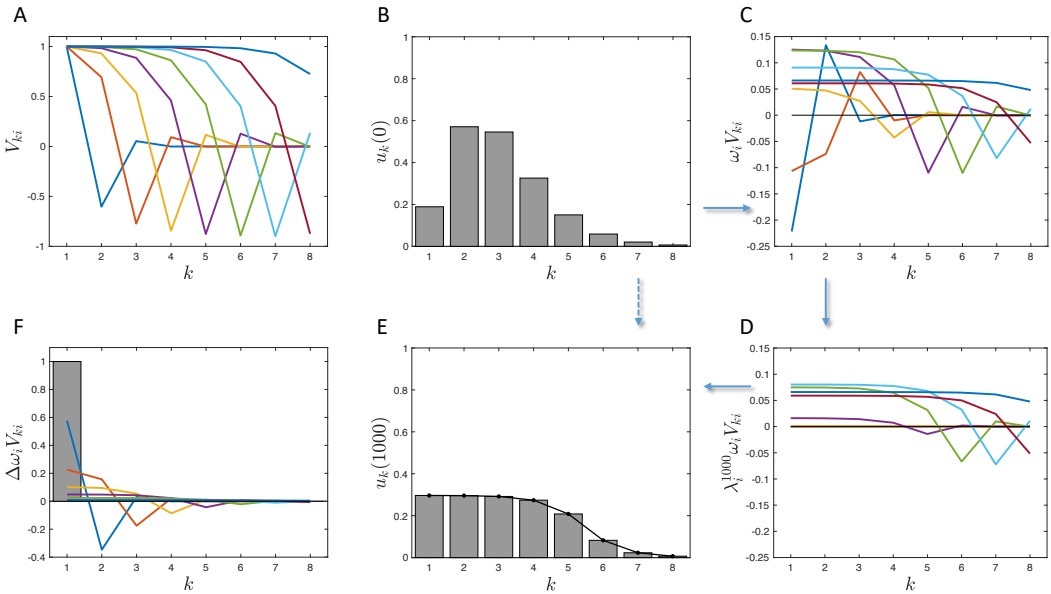

Figure 2: Translation between the model of Benna & Fusi (2016) and the multiscale optimizer works by decomposing the state of the former model into modes, or eigen-patterns of activation that decay independently, which correspond to subweights in the multiscale optimizer. A: All modes for a default Benna-Fusi model with eight variables ($n = 8$). B: An arbitrary initial state of the model. C: Unique eigen-decomposition of the state in Figure 2B. Implied values of the corresponding multiscale optimizer's subweights can be read off as the values of the curves at $k = 1$. D: Decay of the individual modes or subweights for 1000 steps (with no external input) at rates given by their eigenvalues. E: Reconstruction of the final state exactly matches the result of iterating the Benna-Fusi update (dotted arrow from Figure 2B). F: Decomposition of a unit impulse to $u_1$ (e.g., loss gradient, shown as grey bar) as a weighted sum of modes. Learning rates for the corresponding subweights, $\Delta\omega_i$, can be read off as the values of the curves at $k = 1$ (because $V_{1i} = 1$).

## 4    BAYESIAN MULTISCALE OPTIMIZER

We turn now to a normative analysis of learning at multiple timescales, based on Bayesian inference over $1/f$ noise. The Bayesian model introduced here assumes that the latent parameters $\boldsymbol{\theta}$ governing the observed data in some learning task vary over time according to $1/f$ noise. When the statistical model $h(\boldsymbol{x}, \boldsymbol{\theta})$ is linear in $\boldsymbol{\theta}$, exact Bayesian inference is possible with a KF that maintains a posterior over an expanded representation of $\boldsymbol{\theta}$. When the model is nonlinear, approximate Bayesian inference is achieved by an EKF that uses a linear approximation of $h$. We then show that a variational approximation of the KF or EKF, in which the posterior covariance matrix is constrained to be diagonal, yields an extension of the multiscale optimizer that adapts its learning rates online by tracking uncertainty.

### 4.1    GENERATIVE MODEL FOR $1/f$ NOISE

Let $z_i(t)$ be an Ornstein-Uhlenbeck process (i.e., diffusion with decay), with timescale or inverse decay rate $\tau_i$ and diffusion rate $\sigma_i^2$, defined by the following stochastic differential equation:

$$\mathrm{d}z_i = -\tau_i^{-1} z\, \mathrm{d}t + \sigma_i\, \mathrm{d}W. \tag{13}$$

Here $W(t)$ is a standard Wiener process (Brownian motion). As a Gaussian process, $z_i$ has kernel $\mathbb{E}[z_i(t)z_i(t + s)] \propto e^{-|s|/\tau_i}$, implying exponentially decaying autocorrelations. However, a superposition of such processes at different timescales can have qualitatively different properties (Eliazar

& Klafter, 2009). In particular, consider

$$\xi(t) = \sum_{i=1}^{n} z_i(t), \tag{14}$$

where $\tau_i = \nu^i$ and $\sigma_i = \nu^{-i/2}$ for a chosen $\nu > 1$, and $n$ is an integer such that $\tau_n$ is very large. We show in Appendix B that $\xi$ has power-law (i.e., long range) autocorrelations, $\mathbb{E}[\xi(t)\xi(t+s)] \propto |s|^{-1}$ for $s \ll \tau_n$, and accordingly a power spectrum that is well-approximated by $1/f$ for frequencies $f \gg \tau_n^{-1}$. Moreover, $m$ independent copies of this process constitute $m$-dimensional $1/f$ noise, due to the rotational invariance of multidimensional Ornstein-Uhlenbeck processes.

This construction affords a flexible generative model of nonstationarity in a variety of online learning domains, by applying it to the latent parameters governing the relationships among observable variables. Assume we receive observations $\boldsymbol{x}(t), \boldsymbol{y}(t)$ that we wish to model with a statistical model $h$ that is parameterized by $\boldsymbol{\theta} \in \mathbb{R}^m$:

$$\boldsymbol{y}(t) = h(\boldsymbol{x}(t), \boldsymbol{\theta}(t)). \tag{15}$$

For example, $h$ may be a NN with weights $\boldsymbol{\theta}$, input $\boldsymbol{x}$, and target output $\boldsymbol{y}$. The generative side of our Bayesian model posits latent variables $\boldsymbol{z}_i$ ($i = 1, \ldots, n$) such that each $\boldsymbol{z}_i$ is an Ornstein-Uhlenbeck process in $\mathbb{R}^m$ with timescale $\tau_i$, and these processes sum to determine the original parameters:

$$\boldsymbol{\theta}(t) = \sum_{i=1}^{n} \boldsymbol{z}_i(t). \tag{16}$$

These assumptions imply that $\boldsymbol{\theta}$ follows a $1/f$ process, and they entail an expanded state representation, $\boldsymbol{Z} = (\boldsymbol{z}_1, \ldots, \boldsymbol{z}_n) \in \mathbb{R}^{nm}$, that enables efficient inference as described in Section 4.2.

## 4.2 Inference over $1/f$ noise via extended Kalman filter

We consider Bayesian methods that adopt the construction in Section 4.1 as a generative model to account for nonstationarity. Equations 13 and 14 describe a linear dynamic system with state $\boldsymbol{Z} = (z_1, \ldots, z_n) \in \mathbb{R}^n$. If $\xi$ is directly observed at discrete intervals, then optimal Bayesian online prediction of each $\xi(t)$ based on all preceding observations can be implemented by a KF over $\boldsymbol{Z}$ (Kording et al., 2007) (see Appendix D).

We extend this approach to arbitrary statistical models with nonstationarity in their latent parameters, as in Equations 15 and 16. When $h$ is linear in $\boldsymbol{\theta}$ (and hence in $\boldsymbol{Z}$), such as in the regression task and 1-layer perceptron model in Section 5.1, exact inference is possible with a standard KF (Appendix D). For a general $h$, such as a multilayer NN, we use an EKF. The EKF makes a local linear approximation of $h$ based on its Jacobian, the matrix of gradients of predictions $\hat{\boldsymbol{y}}$ with respect to $\boldsymbol{\theta}$ (Appendix E). We use Ollivier's (2018) generalization of the EKF that replaces Gaussian observation noise with any exponential family $p(\boldsymbol{y}|\hat{\boldsymbol{y}})$, which is better suited for modeling discrete outcomes such as the classification tasks of Sections 5.2 and 5.3.

## 4.3 Variational approximation

Finally, we derive a variational approximation of the EKF that extends the multiscale optimizer and affords efficient implementation in large NNs (Appendix F). As is standard, the EKF maintains an iterative prior over the latent state based on all previous observations:

$$p\left(\boldsymbol{Z}(t)|\boldsymbol{x}_{<t}, \boldsymbol{y}_{<t}\right) \sim \mathcal{N}(\boldsymbol{\omega}(t), \boldsymbol{S}(t)). \tag{17}$$

The mean, $\boldsymbol{\omega}(t)$, is the vector of current subweight estimates in the network, while $\boldsymbol{S}(t)$ captures their joint uncertainty and hence determines updates (as a preconditioner on the gradient).

We use variational inference to approximate the distribution in Equation 17 by one in which $\boldsymbol{S}(t) \approx \tilde{\boldsymbol{S}}(t)$, where $\tilde{\boldsymbol{S}}(t)$ is constrained to be diagonal, written as $\tilde{\boldsymbol{S}}(t) = \mathrm{diag}(\boldsymbol{s}^2(t))$. This reduces the complexity from $\mathcal{O}(m^2 n^2)$ to $\mathcal{O}(mnk)$ (the size of the Jacobian, where $k$ is the size of the output layer). The simplest case is a KF that tracks a single $1/f$ variable, with no inputs or latent variables. That is, $y(t) = \xi(t) \in \mathbb{R}^1$ as in Equation 14. Appendix F.1 derives the variational update rule as

$$\omega_i(t+1) = e^{-1/\tau_i}\omega_i(t) + \alpha_i(t)\left(y_i(t) - \hat{y}_i(t)\right), \tag{18}$$

where $y_i(t) - \hat{y}_i(t) = -\partial_w \mathcal{L}$ (i.e., square loss), and the learning rates are given by

$$\alpha_i(t) = \frac{e^{-1/\tau_i} s_i^2(t)}{\sum_{i'} s_{i'}^2(t)}. \tag{19}$$

For the EKF with a general nonlinear model $h(\boldsymbol{x}, \boldsymbol{\theta})$, a slight extension of the variational approximation, derived in Appendix F.3, provides the following update:

$$\omega_{ij}(t+1) = e^{-1/\tau_i} \omega_{ij}(t) - e^{-1/\tau_i} s_{ij}'^2(t) \partial_{w_j(t)} \tilde{\mathcal{L}}. \tag{20}$$

Here $\mathrm{diag}(\boldsymbol{s}'^2)$ is the diagonal variational approximation of the posterior variance after observing $\boldsymbol{y}(t)$, and $\tilde{\mathcal{L}}$ is the negative loglikelihood in the EKF's Gaussian approximate output distribution. Importantly, the update rule for the variance uses the diagonal of the precision matrix but can be calculated without matrix inversion, which is relevant to scaling up to large networks.

Thus the variational method amounts to decomposing every weight in the network as a sum of subweights, $w_j = \sum_i \omega_{ij}$, that learn independently according to their individual gradients, with decay rates $\tau_i$ and learning rates coming from $\tilde{\boldsymbol{S}}(t)$. This is a special case within the family of multiscale optimizers, with additional machinery to adapt the learning rates based on current uncertainty.

## 5 Simulation tests

### 5.1 Regression task

As a simple demonstration, we created an online linear regression task with 10 features (including a bias term), in which the true weights $\boldsymbol{\beta}$ varied over time according to $1/f$ noise using Equation 16. The outcome was generated as $y = \boldsymbol{x}^\top \boldsymbol{\beta}$ (no noise term was needed because $\boldsymbol{\beta}$ is inherently noisy). The corresponding predictive model is a one-layer perceptron, which we write as $\hat{y} = \boldsymbol{x}^\top \boldsymbol{w}$ to distinguish weight estimates ($\boldsymbol{w}$) from true parameters ($\boldsymbol{\beta}$). We model the data using the perceptron and compare methods for optimization, using square loss, $\mathcal{L} = \frac{1}{2}(y - \hat{y})^2$.

We tested two baseline training methods, representing common heuristi practices with nonstationary data (Ditzler et al., 2015; Parisi et al., 2019). First, we tested a batch learning method that uses a fixed memory horizon $H$. To produce a prediction on step $t$, the batch learner fits the perceptron to trials $t-H$ through $t-1$. Figure 3A shows performance is U-shaped: accuracy suffers with short horizons because of sampling error, but it also suffers from longer horizons because older observations are less valid. Second, we tested SGD, in which the weights are updated once after each step $t$, based on $\boldsymbol{x}(t)$ and $y(t)$. Figure 3B shows performance is best with an intermediate learning rate, which roughly corresponds to assuming the environment changes on a single characteristic timescale (see Appendix C).

As applied to the perceptron, the Bayesian $1/f$ model described in Section 4.2 decomposes the weight for each feature $j$ into subweights, $w_j = \sum_i \omega_{ij}$, and tracks the $\omega_{ij}$ jointly with a KF (generalizing Dayan & Kakade, 2000). The subweights combine to predict the outcome on each step: $\hat{y}(t) = \sum_{ij} x_i(t) \omega_{ij}(t)$. Relative to blind prediction (guessing the overall mean of $y$ on

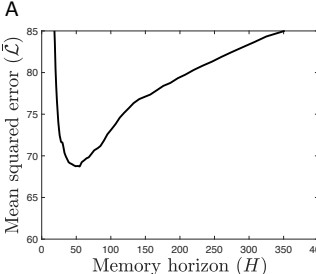
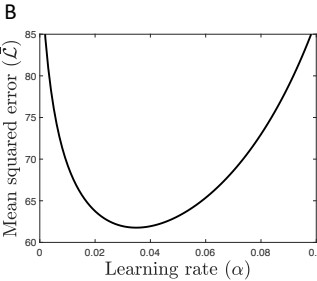
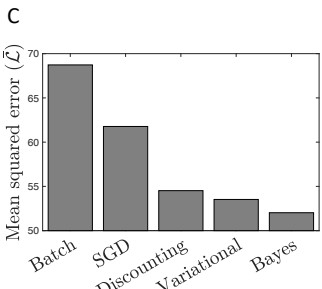

Figure 3: Regression task with $1/f$ dynamics. A: Batch optimization. B: Stochastic gradient descent. C: Model comparison.

every trial), this exact Bayesian solution explains 54.0% more variance in the outcome than the best batch learner, and 25.7% more than the best parameterization of SGD (Figure 3C). Moreover, the variational model that constrains the KF covariance matrix to be diagonal (see Appendix F.2) performs nearly (96.9%) as well as the full Bayesian model.

Finally, we tested a discounting method, similar to windowed batch optimization except all past trials were used for training, discounted by a function of lag. Because the $1/f$ environment has power-law correlations, we weighted each observation $t - k$ by $k^{-a}$ and optimized $a$. This method also significantly outperforms windowed batch and SGD, showing that accounting for an environment's autocorrelation function can achieve much of the advantage of the Bayesian approach. Nevertheless, the full KF and variational methods still outperform discounting, by 5.5% and 2.2% respectively.

## 5.2 LINEAR CLASSIFICATION TASK

Next, we investigated an online 10-way classification task with 10 features (including a bias term). The data were generated by first sampling the class, $y(t) \sim \text{softmax}(\boldsymbol{e}(t))$, and then sampling the feature vector, $\boldsymbol{x}(t)|y(t) \sim \mathcal{N}(\boldsymbol{\mu}_{y(t)}, \boldsymbol{I})$. The logits $e_j$ and the feature-class means $\mu_{ij}$ were independently sampled from $1/f$ processes (using Equation 16), so that there was nonstationarity in both $p(y)$ and $p(\boldsymbol{x}|y)$. Scaling of $\boldsymbol{e}$ and $\boldsymbol{\mu}$ was chosen to equate to maximum possible performance based on perfect knowledge of either one alone (both yielding average loss $\bar{\mathcal{L}} \approx 1$). For the predictive model, we used a one-layer perceptron with a softmax output layer, $\hat{\boldsymbol{y}} = \text{softmax}(\boldsymbol{x}^\top \boldsymbol{W})$, where $\boldsymbol{W}$ is a matrix of learnable feature-class weights. We assumed cross-entropy loss, $\mathcal{L}(y, \hat{\boldsymbol{y}}) = -\log \hat{\boldsymbol{y}}_y$.

The batch method trained the network on trials $t - H$ through $t - 1$ until convergence before predicting $y(t)$. Weight decay was included for regularization, optimized for each value of $H$. Figure 4A shows a U-shaped pattern of performance, reflecting the tradeoff between sampling error and stale data. We also used weight decay with SGD, optimized for each learning rate. Figure 4B again shows a U-shaped pattern of performance. The variational EKF for this task and model is derived in Appendix F.3. We applied $\ell_2$ regularization to the prior on each time step, on par with the SGD and batch optimization methods. Figure 4C shows the variational method outperforms the other two.

## 5.3 MNIST CLASSIFICATION

Finally, we tested our methods on classifying a stream of handwritten MNIST digits (LeCun et al., 2010). We created a nonstationary online learning task by sampling an example from the 10-way MNIST training set on each time step according to class logits that followed $1/f$ noise (Figure 5A). For a predictive model, we used a convolutional neural network (CNN) with two convolution layers followed by two dense layers, with 824458 parameters. This experiment provides a more stringent test of the present methods in several ways. First, it tests whether the EKF's linear approximation and the variational diagonal-variance approximation perform well on a multilayer NN, and whether the algorithm is efficient with a moderately large number of parameters. Second, the predictive model the optimizer is doing inference over is unrelated to how the images and labels were generated. Third, the sampling procedure for $1/f$ noise did not accord with the additive generative process

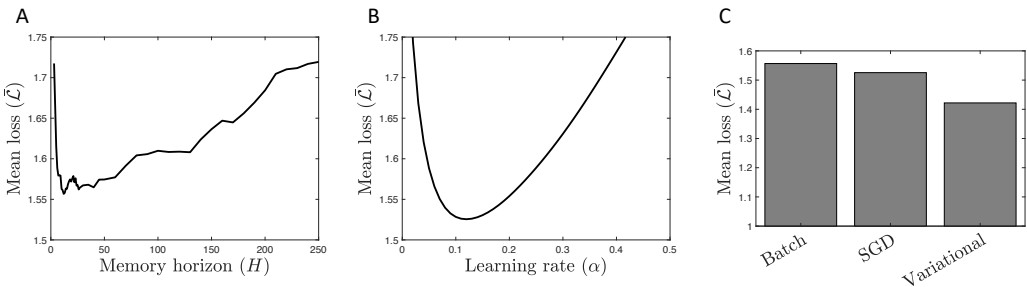

Figure 4: Synthetic classification task with $1/f$ dynamics. Loss is negative log-likelihood. A: Windowed batch optimization. B: Stochastic gradient descent. C: Model comparison.

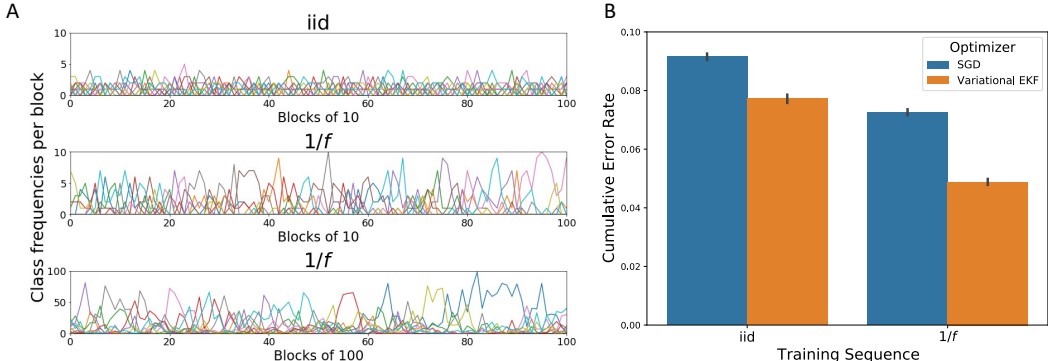

Figure 5: A: Sample frequencies in blocks of consecutive time steps for all 10 MNIST classes (indicated with a unique color per class). The $1/f$ sequence exhibits long-range autocorrelations, with nearly the same pattern over blocks of 10 or 100. B: Model performance over a sequence of 10000 examples (1/6 of MNIST training set).

assumed by the Bayesian model (Section 4.1), but instead used a spectral procedure described in Appendix G.

The variational EKF was compared to SGD with momentum, in both $1/f$ and standard iid environments. Hyperparameters for both methods (noise variance for EKF, learning and momentum rates for SGD) were optimized separately for the two environments. Because the models learn quickly, we evaluated them (within each replication) on a sequence comprising only a random subset of the MNIST training set. Performance was measured by top-1 error rate. This should be interpreted as generalization (i.e., test) performance, because each item was observed only once. The variational EKF outperforms SGD by 1.4% in the iid environment, because its tracking of uncertainty enables more effective gradient steps. However, its advantage jumps to 2.4% in the $1/f$ environment, showing once again that it is better able to leverage dynamics at multiple scales.

## 6 CONCLUSIONS

Our analytic and simulation results demonstrate how online learning performance in nonstationary environments can be improved by incorporating a model of temporal structure. The Bayesian $1/f$ model amounts to distributing knowledge across multiple timescales, and the variational EKF enables approximate implementation in a neural network using subweights with different learning and decay rates. The variational EKF extends the multiscale optimizer, which is closely related to previous models in both neuroscience and ML, and in some cases is equivalent to them despite being simpler in having no coupling between timescales.

We have implemented the variational EKF optimizer in JAX in a format compatible with Optax. In the MNIST simulations of Section 5.3, we find our optimizer code (with 8 timescales) is actually 1.6% faster than Optax's off-the-shelf SGD, in compute time per example. Note also that the multiplexing of subweights is not expensive relative to current optimizers (e.g., Adam; Kingma & Ba, 2015), which also store multiple variables for each weight.

In sum, the multiscale optimizer and variational EKF enjoy a combination of normative, heuristic, and biological justification, good performance, and computational efficiency. Our ongoing work aims to extend the theory in several ways. Chang et al. (2022) compare the present variational method to the fully-decoupled EKF of Puskorius & Feldkamp (2003). Another possible variational method is to assume a block-diagonal matrix that maintains covariance information only between subweights (timescales) within each weight, so that computational complexity still scales linearly with network size. Finally, the present method is not limited to $1/f$ noise but generalizes to other power laws (e.g., $1/f^\beta$) by appropriate choice of the timescales $\tau_i$ and noise variances $\sigma_i$ in the generative model (see Appendix B). If, for example, data or theory is available bearing on the power spectrum of the dynamics in a given domain, the optimizer could be tuned accordingly.

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

## A  DIAGONALIZATION OF BENNA-FUSI TRANSITION MATRIX

This section provides details on diagonalizing the transition matrix of the update rule for the Benna & Fusi (2016) model, and the corresponding reparameterization in terms of the eigenbasis.

First, reparameterize the Benna-Fusi model so that its transition matrix is symmetric, as follows. Recursively define

$$
b_k = \begin{cases} 1 & k = 1 \\ \frac{b_{k-1}c_k}{c_{k-1}} & 1 < k \le n \end{cases}
\tag{21}
$$

and write the state of the Benna-Fusi synapse as

$$
\boldsymbol{\phi} = (\sqrt{b_k}u_k)_{1 \le k \le n}.
\tag{22}
$$

The update becomes

$$
\boldsymbol{\phi}(t+1) = \boldsymbol{\Gamma}\boldsymbol{\phi}(t) + \boldsymbol{d}(t)
\tag{23}
$$

with

$$
\Gamma_{k,k} = 1 - \frac{g_{k-1} + g_k}{C_k}
\tag{24}
$$

$$
\Gamma_{k-1,k} = \Gamma_{k,k-1} = \frac{g_{k-1}}{\sqrt{C_{k-1}C_k}}.
\tag{25}
$$

Symmetry of $\boldsymbol{\Gamma}$ implies it has an orthonormal eigenbasis, $\{\boldsymbol{\psi}_1, \ldots, \boldsymbol{\psi}_n\}$, with corresponding eigenvalues $\lambda_1, \ldots, \lambda_n$. Because the scaling of eigenvectors is arbitrary, we can enforce $\psi_{i,1} = 1$ for all $i$.

To translate the eigenbasis back to $\boldsymbol{u}$, define $\boldsymbol{B} = \mathrm{diag}(\sqrt{\boldsymbol{b}})$ so that $\boldsymbol{\phi} = \boldsymbol{B}\boldsymbol{u}$ and $\boldsymbol{T} = \boldsymbol{B}^{-1}\boldsymbol{\Gamma}\boldsymbol{B}$. It is then easily verified that

$$
\boldsymbol{V}_{\cdot i} = \boldsymbol{B}^{-1}\boldsymbol{\psi}_i,
\tag{26}
$$

is an eigenvector of $\boldsymbol{T}$ with eigenvalue $\lambda_i$. Therefore $\boldsymbol{T} = \boldsymbol{V}\boldsymbol{\Lambda}\boldsymbol{V}^{-1}$, as claimed in the main text. Note that the choice $\psi_{i,1} = 1$ entails $V_{1,i} = 1$ (because $b_1 = 1$). Finally, $\boldsymbol{\Psi} = [\boldsymbol{\psi}_1, \ldots, \boldsymbol{\psi}_n]$ is invertible because it comprises a basis, and therefore so is $\boldsymbol{V} = \boldsymbol{B}^{-1}\boldsymbol{\Psi}$. Therefore the reparameterization $\boldsymbol{u} \mapsto \boldsymbol{\omega} = \boldsymbol{V}^{-1}\boldsymbol{u}$ is well-defined.

## B  GENERATIVE MODEL FOR $1/f$ NOISE

Consider a single Ornstein-Uhlenbeck (OU) process $z(t)$ described by Equation 13 with $\sigma = 1$. The covariance function of $z$ is given by

$$
\mathbb{E}[z(t)z(t+s)] = \int_{-\infty}^{t} e^{-(t-t')/\tau} e^{-(t+s-t')/\tau} \mathrm{d}t'
\tag{27}
$$

$$
= \frac{\tau}{2} e^{-s/\tau}.
\tag{28}
$$

Note that this expression decays exponentially with the lag $s$, yielding short-range autocorrelations. The power spectrum of $z$, as a function of frequency $f$, is the Fourier transform of the covariance function:

$$
P_z(f) = \int_{\mathbb{R}} \frac{\tau}{2} e^{-|s|/\tau} e^{-2\pi i f s} \mathrm{d}s
\tag{29}
$$

$$
= \frac{1}{\tau^{-2} + (2\pi f)^2}
\tag{30}
$$

where $2\pi f$ is angular frequency.

To define a generative model of $1/f$ noise, we define a continuous mixture model over a family of OU processes $(z_\tau)_{0 < \tau \le T}$ for some large $T$:

$$
\xi(t) = \int_0^T 2\tau^{-1} z_\tau(t) \mathrm{d}\tau.
\tag{31}
$$

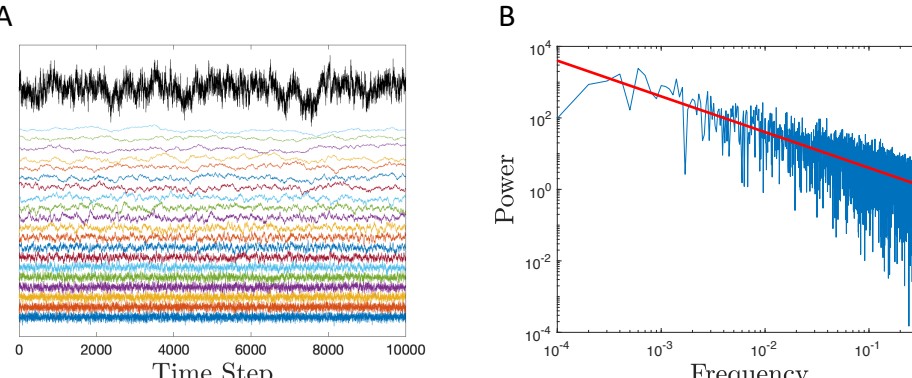

Figure 6: A: Construction of $1/f$ noise (black trajectory at top) as a sum of OU processes with different time constants (colored trajectories). Vertical offsets are applied as a visual aid for discriminating the curves. B: The power spectrum of the aggregate trajectory in log-log coordinates. The red line has unit slope.

The power spectrum of $\xi$ is then a mixture over the component spectra $P_{z_\tau}$:

$$P_\xi(f) = \int_0^T 4\tau^{-2} P_{z_\tau}(f)\mathrm{d}\tau \tag{32}$$

$$= \int_0^T \frac{4\tau^{-2}}{\tau^{-2} + (2\pi f)^2}\mathrm{d}\tau \tag{33}$$

$$= \frac{2}{\pi f}\tan^{-1}(2\pi fT) \tag{34}$$

which is approximately $1/f$ for $f \gg 1/T$, i.e. for all but very low frequencies.

We next define a discrete approximation of the continuum mixture model in Equation 31,

$$\xi(t) = \sum_i z_i(t), \tag{35}$$

where $z_i$ has timescale $\tau_i$ and scaling parameter $\sigma_i$. To approximate a $1/f$ spectrum, $\sigma_i^2(\tau_{i+1}-\tau_i)^{-1}$ should scale as $4\tau_i^{-2}$, the squared weight density in Equation 31 (because power is additive and proportional to $\sigma^2$). For example, the $\tau_i$ could be arithmetically spaced with $\sigma_i \propto \tau^{-1}$. Instead, we assume geometric spacing, with $n$ components defined by $\tau_i = \nu^i$ and $\sigma_i = 2\rho\tau_i^{-1/2}$, where $\rho^2$ is a hyperparameter determining overall steady-state variance. Figure 6 illustrates this construction and exemplifies the accuracy of the discrete approximation.

To define a generative model of $1/f$ noise in $\mathbb{R}^m$, we assume an independent copy of this process for each dimension. The resulting joint process is rotationally symmetric, a property inherited from OU processes. That is, any linear combination $\sum_j c_j\xi_j$ is a 1d $1/f$ process.

## C  KALMAN FILTER FOR A SINGLE TIMESCALE

Start as in Appendix B with a single OU process $z$, with $\sigma = 1$, and assume it is observed at unit intervals $(y_t)_{t\in\mathbb{N}}$ with Gaussian observation noise of variance $\eta^2$. Assume a conjugate iterative prior:

$$z(t)|\boldsymbol{y}_{<t} \sim \mathcal{N}\left(w(t), s^2(t)\right). \tag{36}$$

The posterior after each observation is

$$z(t)|\boldsymbol{y}_{\leq t} \sim \mathcal{N}\left(\frac{\eta^2 w(t) + s^2(t)y(t)}{s^2(t) + \eta^2}, \frac{s^2(t)\eta^2}{s^2(t) + \eta^2}\right). \tag{37}$$

Evolving the process to time $t + 1$ amounts to decay by $e^{-1/\tau}$ and accumulation of new noise. At each time $t' \in [t, t+1]$, variance from the noise appearing at time $t'$ (i.e., from $dW(t')$ in Equation 13) decays by a factor $e^{-2(t+1-t')/\tau}$ by time $t + 1$. Therefore the total accumulated variance is:

$$\int_t^{t+1} e^{-2(t+1-t')/\tau} dt' = \frac{\tau}{2}\left(1 - e^{-2/\tau}\right). \tag{38}$$

Therefore the prior for the next observation is

$$z(t+1)|\boldsymbol{y}_{\leq t} \sim \mathcal{N}\left(e^{-1/\tau}\frac{\eta^2 w(t) + s^2(t)y(t)}{s^2(t) + \eta^2}, e^{-2/\tau}\frac{s^2(t)\eta^2}{s^2(t)+\eta^2} + \frac{\tau}{2}\left(1 - e^{-2/\tau}\right)\right) \tag{39}$$

$$= \mathcal{N}\left(w(t+1), s^2(t+1)\right). \tag{40}$$

We thus obtain the recursion

$$w(t+1) = e^{-1/\tau}\frac{\eta^2 w(t) + s^2(t)y(t)}{s^2(t) + \eta^2} \tag{41}$$

$$= e^{-1/\tau}\left(w(t) + \alpha(t)(y(t) - w(t))\right). \tag{42}$$

This is a temporal-difference learning rule, or gradient descent on squared error $\frac{1}{2}(y(t) - w(t))$, with learning rate $\alpha(t) = \frac{s^2(t)}{s^2(t)+\eta^2}$. This connection between temporal-difference learning and the Kalman filter is well known (Sutton, 1992).

The steady state for $s^2$ is given by the solution to $s^2(t) = s^2(t+1)$, a quadratic with one positive root. If $s^2(0)$ is initialized to this value, then $\alpha(t)$ will be constant. Note that this algorithm (and the extension to $1/f$ noise in Appendix D) generalizes to irregularly spaced observations, in which case one can derive how the optimal learning rate should vary on each step.

## D  KALMAN FILTER OVER $1/f$ NOISE

Assume we receive a sequence of observations at unit time intervals, $y(t)$ for $t \in \mathbb{N}$, and we want to do online prediction under the assumption that $y$ is a $1/f$ noise process. Then we can make the generative assumption $y(t) = \xi(t)$ with $\xi$ defined as at the end of Appendix B. For simplicity and in contrast to Appendix C, we assume no observation noise, because the shortest timescales ($z_1$, etc.) already play this role.

Because $y(t)$ is conditionally independent of the history given (and indeed is fully determined by) the joint state $\boldsymbol{Z}(t) = (z_1(t), \ldots, z_n(t))$, it suffices to compute the posterior for the latter. Write the iterative prior for $\boldsymbol{Z}$ as

$$\boldsymbol{Z}(t)|\boldsymbol{y}_{<t} \sim \mathcal{N}\left(\boldsymbol{\omega}(t), \boldsymbol{S}(t)\right), \tag{43}$$

implying an optimal (maximum-likelihood or least-squares) prediction of $\hat{y}(t) = \sum_i \omega_i(t)$. The posterior after observing $y(t)$ is the intersection of the prior with the hyperplane $\sum_i z_i(t) = y(t)$:

$$\boldsymbol{Z}(t)|\boldsymbol{y}_{\leq t} \sim \mathcal{N}\left(\boldsymbol{\omega}(t) + \frac{\boldsymbol{S}(t)\boldsymbol{1}}{\boldsymbol{1}^\top\boldsymbol{S}(t)\boldsymbol{1}}(y(t) - \hat{y}(t)), \left(\boldsymbol{P}\boldsymbol{S}(t)^{-1}\boldsymbol{P} + \boldsymbol{1}\boldsymbol{1}^\top\right)^{-1}\boldsymbol{P}\right). \tag{44}$$

Here, $\boldsymbol{1}$ is the vector with all elements equal to 1, and $\boldsymbol{P} = \boldsymbol{I} - \frac{1}{n}\boldsymbol{1}\boldsymbol{1}^\top$ is the orthogonal projector.

The prior for the next time step is then obtained by applying decay and adding variance from the noise accumulated over the intervening interval. Define $\boldsymbol{D}$ as the diagonal matrix of decay factors, $e^{-1/\tau_i}$, and $\boldsymbol{N}$ as the diagonal matrix of added noise $2\rho^2(1 - e^{-2/\tau_i})$, obtained by multiplying the RHS of Equation 38 by $\sigma_i^2 = 4\rho^2\tau_i^{-1}$ (from Appendix B). Then we have the update equations for the exact Bayesian model:

$$\boldsymbol{\omega}(t+1) = \boldsymbol{D}\left(\boldsymbol{\omega}(t) + \frac{\boldsymbol{S}(t)\boldsymbol{1}}{\boldsymbol{1}^\top\boldsymbol{S}(t)\boldsymbol{1}}(y(t) - \hat{y}(t))\right) \tag{45}$$

and

$$\boldsymbol{S}(t+1) = \boldsymbol{D}\left(\boldsymbol{P}\boldsymbol{S}(t)^{-1}\boldsymbol{P} + \boldsymbol{1}\boldsymbol{1}^\top\right)^{-1}\boldsymbol{P}\boldsymbol{D} + \boldsymbol{N}. \tag{46}$$

Next, consider the perceptron in Section 5.1, where $y = \boldsymbol{x}^\top \boldsymbol{\theta}$. The $1/f$ generative model assumes $\theta_j = \sum_i z_{ij}$ for each $j$, where $i$ indexes timescales and $j$ indexes features, and $z_{ij}$ has timescale $\tau_i$ and scaling parameter $\sigma_i$ defined as above. The latent state is described by $\boldsymbol{Z} = (z_{ij})_{ij}$. We treat $ij$ as a single composite index, so that $\boldsymbol{Z}$ is a vector. As above, write the iterative prior as

$$\boldsymbol{Z}(t)|\boldsymbol{x}_{<t}, \boldsymbol{y}_{<t} \sim \mathcal{N}\left(\boldsymbol{\omega}(t), \boldsymbol{S}(t)\right). \tag{47}$$

Let $\boldsymbol{X}$ be a multiplexed copy of the input $\boldsymbol{x}$, so that $\boldsymbol{X}_{ij} = \boldsymbol{x}_j$. Assuming square loss, the optimal prediction for $y(t)$ is $\hat{y}(t) = \boldsymbol{X}(t)^\top \boldsymbol{\omega}(t)$. The posterior after observing $y(t)$ is the intersection of the prior with the hyperplane $\boldsymbol{X}(t)^\top \boldsymbol{Z}(t) = y(t)$:

$$\boldsymbol{Z}(t)|\boldsymbol{x}_{\leq t}, \boldsymbol{y}_{\leq t} \sim \mathcal{N}\left(\boldsymbol{\omega}(t) + \frac{\boldsymbol{S}(t)\boldsymbol{X}(t)}{\boldsymbol{X}(t)^\top \boldsymbol{S}(t)\boldsymbol{X}(t)}(y(t) - \hat{y}(t)),\right.$$
$$\left.\left(\boldsymbol{P}_{\boldsymbol{X}(t)}\boldsymbol{S}(t)^{-1}\boldsymbol{P}_{\boldsymbol{X}(t)} + \boldsymbol{X}(t)\boldsymbol{X}(t)^\top\right)^{-1}\boldsymbol{P}_{\boldsymbol{X}(t)}\right), \tag{48}$$

where $\boldsymbol{P}_{\boldsymbol{X}} = \boldsymbol{I} - \boldsymbol{X}\boldsymbol{X}^\top / \boldsymbol{X}^\top \boldsymbol{X}$ is the projector orthogonal to $\boldsymbol{X}$.

Generalizing the definitions above, let $\boldsymbol{D}$ and $\boldsymbol{N}$ be the diagonal matrices of decay factors and noise accumulation, $D_{ij,ij} = e^{-1/\tau_i}$ and $N_{ij,ij} = 2\rho^2(1 - e^{-2/\tau_i})$. Applying these to Equation 48 to obtain the prior for the next time step yields the update equations for the exact Bayesian model of the regression task:

$$\boldsymbol{\omega}(t+1) = \boldsymbol{D}\left(\boldsymbol{\omega}(t) + \frac{\boldsymbol{S}(t)\boldsymbol{X}(t)}{\boldsymbol{X}(t)^\top \boldsymbol{S}(t)\boldsymbol{X}(t)}(y(t) - \hat{y}(t))\right) \tag{49}$$

$$\boldsymbol{S}(t+1) = \boldsymbol{D}\left(\boldsymbol{P}_{\boldsymbol{X}(t)}\boldsymbol{S}^{-1}(t)\boldsymbol{P}_{\boldsymbol{X}(t)} + \boldsymbol{X}(t)\boldsymbol{X}(t)^\top\right)^{-1}\boldsymbol{P}_{\boldsymbol{X}(t)}\boldsymbol{D} + \boldsymbol{N}. \tag{50}$$

Equation 49 exemplifies how the variance matrix, $\boldsymbol{S}(t)$, can be thought of as defining a preconditioner of the gradient, $\boldsymbol{X}(t)(y(t) - \hat{y}(t))$.

# E   EXTENDED KALMAN FILTER

Given a nonlinear model $h(\boldsymbol{x}, \boldsymbol{\theta})$, the $1/f$ EKF posits that the optimal parameters follow $1/f$ dynamics according to Equation 16, with expanded latent state $\boldsymbol{Z} = (\boldsymbol{z}_1, \ldots, \boldsymbol{z}_n)^\top$. It is convenient to introduce the expanded model $\tilde{h}$ defined by $\tilde{h}(\boldsymbol{x}, \boldsymbol{Z}) = h(\boldsymbol{x}, \sum_i \boldsymbol{z}_i)$. The $1/f$ EKF maintains an iterative prior over $\boldsymbol{Z}$ as in Equation 47 and updates that prior by linearizing $\tilde{h}$ about $\boldsymbol{Z} = \boldsymbol{\omega}(t)$:

$$\tilde{h}(\boldsymbol{x}(t), \boldsymbol{Z}) \approx \tilde{h}(\boldsymbol{x}(t), \boldsymbol{\omega}(t)) + J_{\tilde{h}}(\boldsymbol{Z} - \boldsymbol{\omega}(t)). \tag{51}$$

Here, $J_{\tilde{h}} = \frac{\partial \hat{\boldsymbol{y}}}{\partial \boldsymbol{Z}}$ is the Jacobian matrix of $\tilde{h}$, evaluated at $\boldsymbol{Z} = \boldsymbol{\omega}(t)$. Note that $J_{\tilde{h}}$ is just $n$ copies of $J_h$.

Following Ollivier (2018), we assume the observation $y$ is governed by some exponential family $P(y|\boldsymbol{\eta}(\hat{\boldsymbol{y}}))$, with vector of sufficient statistics $\boldsymbol{T}(y)$. The model's output $\hat{\boldsymbol{y}} = \tilde{h}(\boldsymbol{x}, \boldsymbol{\omega})$ is taken to encode the predicted mean parameter of that family: $\hat{\boldsymbol{y}} = \mathbb{E}_{y \sim P(\cdot|\boldsymbol{\eta}(\hat{\boldsymbol{y}}))}[\boldsymbol{T}(y)]$ (this can be read as a definition of the mapping $\hat{\boldsymbol{y}} \mapsto \boldsymbol{\eta}(\hat{\boldsymbol{y}})$). It then approximates the conditional distribution of the sufficient statistics as a Gaussian,

$$p(\boldsymbol{T}(y)|\hat{\boldsymbol{y}}) \approx \mathcal{N}\left(\hat{\boldsymbol{y}}, \boldsymbol{R}_{\hat{\boldsymbol{y}}}\right), \tag{52}$$

where $\boldsymbol{R}_{\hat{\boldsymbol{y}}} = \text{Var}\left(\boldsymbol{T}(y)|\hat{\boldsymbol{y}}\right)$ is the conditional variance.

For example, when $h$ is a classification model as in Section 5.2 or 5.3, the output of the network is a vector $\hat{\boldsymbol{y}}$ of class probabilities, and the sufficient statistics $\boldsymbol{T}(y)$ are a one-hot vector. For numerical stability, we exclude the final element of $\hat{\boldsymbol{y}}$ and $\boldsymbol{T}(y)$, which are determined by the other elements. The conditional outcome variance is given by

$$[\boldsymbol{R}_{\hat{\boldsymbol{y}}}]_{i,j} = \begin{cases} \hat{y}_i(1 - \hat{y}_i) & i = j \\ -\hat{y}_i \hat{y}_j & i \neq j. \end{cases} \tag{53}$$

Under the approximations of Equations 51 and 52, the posterior is given by the standard KF formula:

$$\boldsymbol{Z}(t)|\boldsymbol{x}_{\leq t}, \boldsymbol{y}_{\leq t} \sim \mathcal{N}\left(\boldsymbol{\omega}(t) + \boldsymbol{S}(t)\boldsymbol{J}_{\tilde{h}}^{\top}\left(\boldsymbol{J}_{\tilde{h}}\boldsymbol{S}(t)\boldsymbol{J}_{\tilde{h}}^{\top} + \boldsymbol{R}_{\hat{\boldsymbol{y}}(t)}\right)^{-1}\left(\boldsymbol{T}(y(t)) - \hat{\boldsymbol{y}}(t)\right),\right.$$

$$\left.\boldsymbol{S}(t) - \boldsymbol{S}(t)\boldsymbol{J}_{\tilde{h}}^{\top}\left(\boldsymbol{J}_{\tilde{h}}\boldsymbol{S}(t)\boldsymbol{J}_{\tilde{h}}^{\top} + \boldsymbol{R}_{\hat{\boldsymbol{y}}(t)}\right)^{-1}\boldsymbol{J}_{\tilde{h}}\boldsymbol{S}(t)\right). \tag{54}$$

Applying decay ($\boldsymbol{D}$) and accumulated noise ($\boldsymbol{N}$) as in Appendix D to obtain the prior for the next time step yields the update equations for the $1/f$ EKF:

$$\boldsymbol{\omega}(t+1) = \boldsymbol{D}\left(\boldsymbol{\omega}(t) + \boldsymbol{S}(t)\boldsymbol{J}_{\tilde{h}}^{\top}\left(\boldsymbol{J}_{\tilde{h}}\boldsymbol{S}(t)\boldsymbol{J}_{\tilde{h}}^{\top} + \boldsymbol{R}_{\hat{\boldsymbol{y}}(t)}\right)^{-1}\left(\boldsymbol{T}(y(t)) - \hat{\boldsymbol{y}}(t)\right)\right) \tag{55}$$

$$\boldsymbol{S}(t+1) = \boldsymbol{D}\left(\boldsymbol{S}(t) - \boldsymbol{S}(t)\boldsymbol{J}_{\tilde{h}}^{\top}\left(\boldsymbol{J}_{\tilde{h}}\boldsymbol{S}(t)\boldsymbol{J}_{\tilde{h}}^{\top} + \boldsymbol{R}_{\hat{\boldsymbol{y}}(t)}\right)^{-1}\boldsymbol{J}_{\tilde{h}}\boldsymbol{S}(t)\right)\boldsymbol{D} + \boldsymbol{N}. \tag{56}$$

Although we apply the EKF to feedforward NNs in this paper, we note that the approach naturally generalizes to other probabilistic causal models relating the observed variables. Thus it offers a means to model nonstationarity that covers all of the traditional forms of distribution shift, following the causal framework of Schölkopf et al. (2012). Consider a classification task, with input features $\boldsymbol{x}(t) \in \mathbb{R}^m$ and class labels $y(t) \in \mathbb{Z}_k$. Under a generative causal model where $\boldsymbol{x}$ depends on $y$, label shift can be modeled by $1/f$ noise in the distribution $p(y)$, for example $y \sim \text{softmax}(\boldsymbol{\ell})$ with $\boldsymbol{\ell}$ given by Equation 16. Manifestation shift can be modeled by $1/f$ noise in the distributions $p(\boldsymbol{x}|y)$, for example $\boldsymbol{x}(t)|y(t) \sim \mathcal{N}\left(\boldsymbol{\mu}_{y(t)}, \boldsymbol{\Sigma}\right)$ with $\boldsymbol{\mu}_y$ given by Equation 16 for each $y$. Likewise, under a discriminative causal model where $y$ depends on $\boldsymbol{x}$, covariate shift can be modeled by $1/f$ noise in $p(\boldsymbol{x})$, and concept shift can be modeled by $1/f$ noise in $p(y|\boldsymbol{x})$.

## F  VARIATIONAL APPROXIMATION

This section derives variational approximations of the KF and EKF models, with $\boldsymbol{S}(t) \approx \tilde{\boldsymbol{S}}(t)$ where $\tilde{\boldsymbol{S}}(t) := \text{diag}(\boldsymbol{s}^2(t))$. Thus the resulting algorithms need only track the individual variance terms in $\boldsymbol{s}^2(t)$ rather than the full covariance matrix. Moreover, the update equations (63,64,68,69,74,77) all avoid matrix inversion—even though this might initially seem necessary from Equation 57—a property that may be relevant for efficient scaling.

### F.1  UNCUED INFERENCE

We begin with the simple Bayesian $1/f$ model in Equations 45 and 46, where there are no predictors and the model merely tracks an observable $y(t)$. Given an arbitrary Gaussian distribution, the variational approximation (in the sense of minimizing Kullback-Leibler divergence) by another Gaussian with diagonal covariance matrix is obtained by taking the diagonal of the original distribution's precision matrix. Thus in the present case we have

$$s_i^{-2}(t+1) = \left[\boldsymbol{S}^{-1}(t+1)\right]_{i,i} \tag{57}$$

where $\boldsymbol{S}(t+1)$ is given by Equation 46 with $\boldsymbol{S}^{-1}(t)$ replaced by $\tilde{\boldsymbol{S}}^{-1}(t)$ (the inductive assumption). To calculate $\boldsymbol{S}^{-1}(t+1)$ under this assumption, we first observe the identity

$$\left(\boldsymbol{P}\,\text{diag}\left(\boldsymbol{s}^{-2}\right)\boldsymbol{P} + \boldsymbol{1}\boldsymbol{1}^{\top}\right)^{-1}\boldsymbol{P} = \text{diag}\left(\boldsymbol{s}^2\right) - \frac{\boldsymbol{s}^2\boldsymbol{s}^{2\top}}{\sum_i s_i^2}. \tag{58}$$

Therefore Equation 46 becomes

$$\boldsymbol{S}(t+1) = \boldsymbol{D}\tilde{\boldsymbol{S}}(t)\boldsymbol{D} + \boldsymbol{N} - \frac{(\boldsymbol{D}\boldsymbol{s}^2(t))(\boldsymbol{D}\boldsymbol{s}^2(t))^{\top}}{\sum_i s_i^2(t)} \tag{59}$$

$$:= \text{diag}(\boldsymbol{a}) - \frac{\boldsymbol{b}\boldsymbol{b}^{\top}}{c} \tag{60}$$

where $\text{diag}(\boldsymbol{a}) = \boldsymbol{D}\tilde{\boldsymbol{S}}(t)\boldsymbol{D} + \boldsymbol{N}$, $\boldsymbol{b} = \boldsymbol{D}\boldsymbol{s}^2(t)$, and $c = \sum_i s_i^2(t)$. We next use the identity

$$\left(\text{diag}\left(\boldsymbol{a}\right) - \frac{\boldsymbol{b}\boldsymbol{b}^{\top}}{c}\right)^{-1} = \text{diag}\left(\frac{1}{\sqrt{\boldsymbol{a}}}\right)\left(\boldsymbol{I} + \frac{\left(\boldsymbol{a}^{-1/2} \circ \boldsymbol{b}\right)\left(\boldsymbol{a}^{-1/2} \circ \boldsymbol{b}\right)^{\top}}{c - \sum \frac{b_i^2}{a_i}}\right)\text{diag}\left(\frac{1}{\sqrt{\boldsymbol{a}}}\right), \tag{61}$$

implying the diagonal elements are

$$\left[\left(\text{diag}(\boldsymbol{a}) - \frac{\boldsymbol{b}\boldsymbol{b}^\top}{c}\right)^{-1}\right]_{i,i} = \frac{1}{a_i}\left(1 + \frac{\frac{b_i^2}{a_i}}{c - \sum_{i'}\frac{b_{i'}^2}{a_{i'}}}\right). \tag{62}$$

Combining Equations 57, 60, and 62 gives the final form of the variational update:

$$s_i^2(t+1) = \frac{\left(s_i^2(t)e^{-1/\tau_i} + 4\rho^2\sinh\frac{1}{\tau_i}\right)^2\Omega}{\left(s_i^2(t) + 4\rho^2 e^{1/\tau_i}\sinh\frac{1}{\tau_i}\right)\Omega + s_i^4(t)} \tag{63}$$

with

$$\Omega = \sum_i \frac{4\rho^2 s_i^2(t)\sinh\frac{1}{\tau_i}}{e^{-1/\tau_i}s_i^2(t) + 4\rho^2\sinh\frac{1}{\tau_i}}. \tag{64}$$

This update of the variance converges exponentially to a unique fixed point. Numerical simulations confirm that, in this limit, $s_i^2$ is larger for smaller $\tau_i$, meaning faster learning rates for shorter timescales. If the prior variances are initialized at the fixed point then they are constant throughout learning.

By substituting the diagonal matrix $\tilde{\boldsymbol{S}}(t)$ for $\boldsymbol{S}(t)$, the update for the mean in Equation 45 simplifies to

$$\omega_i(t+1) = e^{-1/\tau_i}\omega_i(t) - \alpha_i(t)\left(\hat{y}(t) - y(t)\right) \tag{65}$$

where $\hat{y}(t) - y(t)$ is the loss gradient (assuming square loss), and the learning rates are given by

$$\alpha_i(t) = \frac{e^{-1/\tau_i}s_i^2(t)}{\sum_{i'} s_{i'}^2(t)}. \tag{66}$$

That is, the subweights learn independently according to their gradients, with different decay rates and learning rates. Thus we have recovered an extension of the multiscale optimizer, with an additional mechanism that adapts the learning rates on each time step (via $\boldsymbol{s}$).

## F.2 KALMAN FILTER

To derive the variational KF for the regression task, we apply the analysis of Section F.1 to the KF update in Equations 49 and 50. Paralleling the derivation of Equation 59, the variance update in Equation 50 (i.e., before applying the diagonal variational approximation) can be written as

$$\boldsymbol{S}(t+1) = \boldsymbol{D}\tilde{\boldsymbol{S}}(t)\boldsymbol{D} + \boldsymbol{N} - \frac{(\boldsymbol{D}(\boldsymbol{x}(t)\circ\boldsymbol{s}^2(t)))(\boldsymbol{D}(\boldsymbol{x}(t)\circ\boldsymbol{s}^2(t)))^\top}{\sum_{ij} x_{ij}^2(t)s_{ij}^2(t)}. \tag{67}$$

Paralleling the derivation of Equations 63 and 64, the variational update comes out to be

$$s_{ij}^2(t+1) = \frac{\left(s_{ij}^2(t)e^{-1/\tau_i} + 4\rho^2\sinh\frac{1}{\tau_i}\right)^2\Omega}{\left(s_{ij}^2(t) + 4\rho^2 e^{1/\tau_i}\sinh\frac{1}{\tau_i}\right)\Omega + x_j^2(t)s_{ij}^4(t)} \tag{68}$$

with

$$\Omega = \sum_{ij} \frac{4\rho^2 x_j^2(t)s_{ij}^2(t)\sinh\frac{1}{\tau_i}}{e^{-1/\tau_i}s_{ij}^2(t) + 4\rho^2\sinh\frac{1}{\tau_i}}. \tag{69}$$

By substituting the diagonal matrix $\tilde{\boldsymbol{S}}(t)$ for $\boldsymbol{S}(t)$, the mean update in Equation 49 simplifies to

$$\omega_{ij}(t+1) = e^{-1/\tau_i}\omega_{ij}(t) - \alpha_{ij}x_j(t)\left(\hat{y}(t) - y(t)\right) \tag{70}$$

where $x_j(t)\left(\hat{y}(t) - y(t)\right)$ is the loss gradient for $\omega_{ij}$, and the learning rates are given by

$$\alpha_{ij} = \frac{e^{-1/\tau_i}s_{ij}^2(t)}{\sum_{i'j'} x_{j'}^2(t)s_{i'j'}^2(t)}. \tag{71}$$

Thus we have recovered an extension of the multiscale optimizer, with an additional mechanism that adapts the learning rates on each time step (via $\boldsymbol{x}$ and $\boldsymbol{s}$).

### F.3  EXTENDED KALMAN FILTER

To derive a closed-form variational approximation for the general EKF, such as for the classification models in Sections 5.2 and 5.3, it turns out that we need to apply the variational approximation to the posterior in Equation 54, rather than to the iterative prior in Equation 56. Using Woodbury's identity, the posterior variance can be rewritten as

$$\boldsymbol{S}(t) - \boldsymbol{S}(t)\boldsymbol{J}_{\tilde{h}}^{\top}\left(\boldsymbol{J}_{\tilde{h}}\boldsymbol{S}(t)\boldsymbol{J}_{\tilde{h}}^{\top} + \boldsymbol{R}_{\hat{\boldsymbol{y}}(t)}\right)^{-1}\boldsymbol{J}_{\tilde{h}}\boldsymbol{S}(t) = (\boldsymbol{J}_{\tilde{h}}^{\top}\boldsymbol{R}_{\hat{\boldsymbol{y}}(t)}^{-1}\boldsymbol{J}_{\tilde{h}} + \boldsymbol{S}^{-1}(t))^{-1}. \tag{72}$$

The form on the RHS is convenient because it is in terms of precision, allowing us to read off the diagonal entries directly. That is, the variational approximation for the posterior variance is $\mathrm{diag}(\boldsymbol{s}'^2(t))$, with

$$s_{ij}'^2(t) = \left(\left[\boldsymbol{J}_{\tilde{h}}^{\top}\boldsymbol{R}_{\hat{\boldsymbol{y}}(t)}^{-1}\boldsymbol{J}_{\tilde{h}}\right]_{ij,ij} + s_{ij}^{-2}(t)\right)^{-1}. \tag{73}$$

Here was have used the inductive assumption $\boldsymbol{S}(t) \approx \mathrm{diag}(\boldsymbol{s}^2(t))$. Applying the transition from posterior on step $t$ to prior on step $t+1$, we obtain the variance update for the variational model, replacing Equation 56:

$$s_{ij}^2(t+1) = D_{ij,ij}^2 s_{ij}'^2(t) + N_{ij,ij}. \tag{74}$$

Woodbury's identity also enables the mean update from Equation 55 to be rewritten, as

$$\boldsymbol{\omega}(t+1) = \boldsymbol{D}\left(\boldsymbol{J}_{\tilde{h}}^{\top}\boldsymbol{R}_{\hat{\boldsymbol{y}}(t)}^{-1}\boldsymbol{J}_{\tilde{h}} + \boldsymbol{S}^{-1}(t)\right)^{-1}\left(\boldsymbol{S}^{-1}(t)\boldsymbol{\omega}(t) + \boldsymbol{J}_{\tilde{h}}^{\top}\boldsymbol{R}_{\hat{\boldsymbol{y}}(t)}^{-1}\left(\boldsymbol{T}(y(t)) - \hat{\boldsymbol{y}}(t) + \boldsymbol{J}_{\tilde{h}}\boldsymbol{\omega}(t)\right)\right). \tag{75}$$

Substituting the gradient of the EKF's approximate likelihood (denoted $\tilde{\mathcal{L}}$) from Equation 52,

$$\partial_{\boldsymbol{\omega}(t)}\tilde{\mathcal{L}} = \boldsymbol{J}_{\tilde{h}}^{\top}\boldsymbol{R}_{\hat{\boldsymbol{y}}(t)}^{-1}(\hat{\boldsymbol{y}}(t) - \boldsymbol{T}(y(t))), \tag{76}$$

yields

$$\boldsymbol{\omega}(t+1) = \boldsymbol{D}\left(\boldsymbol{\omega}(t) - \left(\boldsymbol{J}_{\tilde{h}}^{\top}\boldsymbol{R}_{\hat{\boldsymbol{y}}(t)}^{-1}\boldsymbol{J}_{\tilde{h}} + \boldsymbol{S}^{-1}(t)\right)^{-1}\partial_{\boldsymbol{\omega}(t)}\tilde{\mathcal{L}}\right). \tag{77}$$

The preconditioner on the gradient here is the posterior variance (see Equation 72), which we have approximated as $\mathrm{diag}(\boldsymbol{s}'^2(t))$. Although not necessarily entailed by the variational approximation, we can consider applying the same approximation in updating the mean. This yields

$$\omega_{ij}(t+1) = e^{-1/\tau_i}\omega_{ij}(t) - e^{-1/\tau_i}s_{ij}'^2(t)\partial_{w_j(t)}\tilde{\mathcal{L}}, \tag{78}$$

which once again extends the multiscale optimizer by adapting its learning rate to current uncertainty.

## G  IMPLEMENTATION DETAILS

Latent parameters for the synthetic tasks in Sections 5.1 and 5.2 were sampled using the generative model in Appendix B. That is, the data-generating process matched the generative assumptions of the Bayesian model in both of these cases. We used 20 timescales, geometrically spaced from $\tau_1 = 1$ to $\tau_{20} = 1000$, as illustrated in Figure 6A. Each component OU process was run for $10\tau_i$ burn-in steps to ensure stationarity. The regression task was run for 10k trials, and the linear classification task for 1000 trials. The $1/f$ power spectrum was confirmed with a log-log plot (see Figure 6B).

In the regression task of Section 5.1, the first feature was a constant bias term, $x_1 \equiv 1$. The other 9 features were sampled independently as $\mathcal{N}(\boldsymbol{0}, \boldsymbol{I})$ on each time step.

For the linear classification task of Section 5.2, class logits were sampled from $1/f$ processes and then multiplied by 0.886 before entering into softmax to determine class probabilities. Feature-class means were fixed at 1 for the first feature (i.e., bias term) and were sampled from mutually independent $1/f$ processes for features 2-10, multiplied by 0.224. These scaling factors were chosen so that perfect knowledge of either the prior probabilities or the feature-class means on every trial

would yield ideal-observer performance of $\bar{\mathcal{L}} \approx 1$. Perfect knowledge of both would yield $\bar{\mathcal{L}} \approx 0.35$. These were merely guidelines for equating prior and likelihood information, as perfect knowledge of either source of information is not possible even with an optimal model of the dynamics.

In both Sections 5.1 and 5.2, all Bayesian and variational models assumed 10 timescales, geometrically spaced from $\tau_1 = 2$ to $\tau_{10} = 800$. This deliberate deviation from the data-generating process (see the 20 timescales listed above) provided a mild test of robustness, specifically the hypothesis that it is the aggregate $1/f$ character of the environment and of the model that matters, not the choice of component timescales used to approximate that character.

For the MNIST classification task in Section 5.3, the class on each time step was sampled by softmax applied to class logits that varied across steps according to $1/f$ noise. The item on that time step was than sampled without replacement from all members of that class in the MNIST training set. The logit sequences were generated as follows. First, for each class, we sampled a Standard Gaussian random vector of length equal to the total number of time steps (10k). Then we applied a discrete Fourier transform, multiplied the result by $f^{-1/2}$, and finally applied the inverse Fourier transform. Thus the resulting sequence had $1/f$ power spectrum. For the iid environment, the logits were sampled as white noise (constant power spectrum), by sampling a Standard Gaussian vector as above and using it unaltered.

