# OpenReview forum: "Learning in temporally structured environments"
_ICLR.cc/2023/Conference — ICLR 2023 poster_

### Official Review · Reviewer_8p8i · 2022-10-20

**Confidence:** 2
**Correctness:** 4
**Technical Novelty And Significance:** 4
**Empirical Novelty And Significance:** 4
**Recommendation:** 8

**Clarity, Quality, Novelty And Reproducibility:**

- Quality and Clarity: the paper is very well written. It is high quality and very clear. To the best of my knowledge it is also original.
- Reproducibility: The authors have not provided code which would help reproducibility. The experiments are described in fair bit of detail.

**Strength And Weaknesses:**

Strengths:
- The paper is well written, thorough in theory and background information (it does require reading the appendix to fully grasp the derivations).
- Simplifying and unifying various timescale learning methods into a single framework is a significant contribution.
- I specially enjoyed figures 1 and 2. They do a great job of summarizing the whole paper in to two clear, concise and simple figures + captions.

Weaknesses:
- I'm hard pressed to find one. I understand this is a mostly theoretical paper but it would still have been great to see experiments on more than the simulated cases.

**Summary Of The Paper:**

The paper proposes a generalized multiscale learning framework for learning in temporally structured environments. The basic idea is that each weight in a NN is decomposed into a sum of subweights, each updated with different learning and decay rates, with all components evolving independently as compared to previous models which involved complex coupling between timescales. They show that these previous models can be shown to be equivalent to the multiscale learner, with no need for these couplings across timescales. They also show that momentum learning can be characterized within the multiscale learner framework as well. Lastly they derive an equivalent model of bayesian inference over 1/f noise and show that that model is also a special case of the multiscale learner framework.

**Summary Of The Review:**

I believe this is a paper worth appearing at ICLR. It presents a well thought out and well presented unifying and simplifying framework for learning in temporally structured environments.

---

> ### Author Response · Authors · 2022-11-18
> **Reply to Reviewer 8p8i**
>
> Thank you for the positive feedback and appreciation of our theoretical contributions. The suggestions in this review align with ones by the other reviewers and have led to important improvements in the revision.
>
> > It does require reading the appendix to fully grasp the derivations
>
> We have done some fine-tuning to help the main text stand on its own better (see various marked changes in Sections 3-4). The larger revisions we made to better separate the generative side of the Bayesian model from the inferential side, and the generative model from the data sampling in the experiments, should also help (see items 2-3 in our joint response to all reviewers). Nevertheless, we agree the appendices are still essential for a full technical understanding.
>
> > I understand this is a mostly theoretical paper but it would still have been great to see experiments on more than the simulated cases.
>
> We agree, and we have added a third simulation using real data (MNIST with nonstationary label sequences) with a significantly larger CNN as the predictive model. Please see Section 5.3 and the detailed summary in our joint response to all reviewers (item 1). We believe this addition rounds out the paper and complements our strong theoretical contributions.
>
> > The authors have not provided code which would help reproducibility.
>
> We agree, and we plan to add a github link to the final de-anonymized version (if accepted), including the code for the three experiments and Optax-style code for the optimizer that other researchers can use in their models with minimal effort.

---

> > ### Comment · Reviewer_8p8i · 2022-11-18
> > **Response to authors**
> >
> > Thank you for your response and making appropriate changes to the paper.

---

### Official Review · Reviewer_6H7B · 2022-10-24

**Confidence:** 3
**Correctness:** 4
**Technical Novelty And Significance:** 3
**Empirical Novelty And Significance:** 2
**Recommendation:** 6

**Clarity, Quality, Novelty And Reproducibility:**

**Clarity**

The paper is quite well written and easy to follow. Except for a few instances referring to a concurrent submission, the paper is self-contained with the relevant details described in sufficient depth. Minor correction: In the last paragraph of Section 4.1 line 5, "manifestation shift" -> "Manifestation shift"


**Quality and Novelty**

While the quality of the contributions in the paper is high, the perspective is not particularly novel. Multiscale learning has been explored in the RNN literature in the past (which is not mentioned in the paper) - e.g. [1]. Nonetheless, the insights presented are a substantial contribution to the community.


**Reproducibility**
The authors did not provide code with their submission, but the experiments are small and straight-forward, accompanied with enough details for reproduction. The theoretical results also stated and derived with clarity,


[1] Long Expressive Memory for Sequence Modeling, ICLR 2022

**Strength And Weaknesses:**

**Strengths**:

- The paper tackles an important problem of learning in non-i.i.d settings, in the specific case where multiple learning processes operate at different timescales. The authors present a unifying framework of multiscale optimizers for learning in such settings. The framework encompasses well studied approaches of momentum learning and the Benna-Fusi model, providing novel insights for momentum learning.
- The Bayesian formulation provides a simple and straightforward approach for implementing the multiscale optimizer.
- The synthetic simulations provide _some_ evidence on the applicability of the multiscale optimizer in practical settings. (I discuss this a bit more in the weakness)

**Weaknesses**

- The paper studies a particular setting where the multi-scale weights are additive. This however seems like a somewhat limiting assumption to me. The authors do not discuss this choice in the paper and it is not clear whether this assumption holds true in realistic settings.
- While the simulations provide a sanity check for the approach, they leave quite a bit to be desired. As mentioned in the first point, the assumption of weights of the weights of the different timescales being additive might not hold in any realistic setting. Experiments on more realistic settings would be helpful to address this concern.

**Summary Of The Paper:**

Temporally structured environments involve multiple learning processes operating at different time-scales. The paper provides a unifying view of learning in these temporally structured environments. The authors begin by formalizing the multi-scale learning setup, where the weights are decomposed into sub-weights, each characterized by a timescale and learning rate and decay. The authors first establish fast weights as an instantiation of the multiscale optimizer with 2 timescales. Next, the authors establish the equivalence between the multiscale optimizer (in the configuration of a fast-weight model) and the Benna-Fusi model for real synapses, as well as momentum learning. An interesting observation from the analysis is that momentum can be characterized as a fast weight with a negative learning rate. The authors then present a Bayesian inference view for a multiscale learner, adopting the $1/f$ model, and describe exact inference with a Kalman filter and approximate inference with extended Kalman filters and variational approximation for extended Kalman filter. Finally the authors validate the multiscale learning framework with some experiments on synthetic tasks.

**Summary Of The Review:**

To summarize, the paper presents a unifying perspective for multiscale optimization in some non-iid settings. The paper makes useful connections to existing ideas and proposes a novel Bayesian formulation for multiscale optimizers. The empirical evaluations seem lacking given certain assumptions made in the formalization. More realistic experiments would be helpful in addressing this shortcoming. Nonetheless, the conceptual and theoretical contributions of the paper are substantial, so I lean towards acceptance.

---

> ### Author Response · Authors · 2022-11-18
> **Reply to Reviewer 6H7B (part 1 of 2)**
>
> Thank you for the overall positive assessment and constructive comments. We have addressed all five of your suggestions, as detailed below. We believe the paper is significantly clearer and stronger as a result.
>
> > The paper studies a particular setting where the multi-scale weights are additive. This however seems like a somewhat limiting assumption to me. The authors do not discuss this choice in the paper and it is not clear whether this assumption holds true in realistic settings.
>
> To answer this, let us first distinguish actual statistics of the task environment (in our experiments and in potential applications) from generative assumptions of the Bayesian model. As we discuss in our joint response to all reviewers (item 2), we have made this distinction clearer in the revision. Briefly, our Bayesian approach represents multiscale nonstationarity with an additive generative model, which posits latent variables (OU processes) that sum to the true parameters: $\boldsymbol{\theta} = \sum \boldsymbol{z}_i$ (Eq 16). These generative assumptions have two useful mathematical properties: (a) they imply $\boldsymbol{\theta}$ is a $1/f$ process and (b) they admit efficient inference over the joint state $\boldsymbol{Z}$. As is common with Bayesian models, the latent $\boldsymbol{z}_i$ variables exist only inside the model and need not correspond to anything real in the environment.
>
> With this backdrop, the comment in the review can be broken into two questions: (1) How well do the present methods perform when the environment is not well-described by the generative model? (2) How limiting is the generative assumption that the latent $\boldsymbol{z}_i$ variables combine additively?
>
> Concerning question 1, as we summarize at the end of Section 1, we test robustness by using a different set of timescales for task generation versus learning (Sections 5.1-5.2) and a construction of $1/f$ noise that is different from the additive process that the model assumes (Section 5.3). The latter is most directly related to your question. In short, the nonstationarity in the MNIST label sequence was created not as a sum of OU processes at different time scales but by directly manipulating the discrete Fourier transform, as described at the end of Appendix G. The superior performance of the variational EKF in that environment indicates it is not limited to settings defined by additive processes.
>
> Concerning question 2, as we briefly mention at the end of Section 6 when discussing directions for ongoing work, the generative assumption of additive processes is not limited to $1/f$ noise. By modifying the mixture coefficients in the generative model (see Eq 31), we can obtain alternative power spectra for the aggregate process (paralleling the derivation in Eqs 32-34). The mixture coefficients determine the noise variance $\sigma^2_i$ for each component process. These in turn determine how posterior variance is updated, and hence how learning rates adapt, in the inferential model. Thus, as we write in Section 6, if data or theory were available bearing on the power spectrum of the dynamics in a given domain, the optimizer could be tuned accordingly.
>
> > While the simulations provide a sanity check for the approach, they leave quite a bit to be desired. As mentioned in the first point, the assumption of weights of the weights of the different timescales being additive might not hold in any realistic setting. Experiments on more realistic settings would be helpful to address this concern.
>
> Although this is primarily a theory paper and we believe the analytic results make a strong contribution on their own, we agree the simulations in the previous draft were not a particularly stringent test. Therefore, we have added a third simulation using real data (MNIST with nonstationary label sequences) with a significantly larger CNN as the predictive model. Please see Section 5.3 and the detailed summary in our joint response to all reviewers (item 1). Also, as noted in our response to the previous comment, the alternative construction of $1/f$ noise in this simulation supports the conclusion that the generative assumption of additive processes does not limit the model’s applicability.
>
> > Minor correction: In the last paragraph of Section 4.1 line 5, "manifestation shift" -> "Manifestation shift”
>
> Fixed (moved to Appendix E)

---

> > ### Author Response · Authors · 2022-11-18
> > **Reply to Reviewer 6H7B (part 2 of 2)**
> >
> > > While the quality of the contributions in the paper is high, the perspective is not particularly novel. Multiscale learning has been explored in the RNN literature in the past (which is not mentioned in the paper) - e.g. [1]. Nonetheless, the insights presented are a substantial contribution to the community.
> >
> > Thanks for pointing us to the LEM paper, which we now mention in Section 1. There are clearly connections to our paper that will be profitable to explore in detail. Two obvious questions are whether the ODEs underlying LEM could be derived from a similar Bayesian analysis positing correlations at multiple time scales, and whether the coupling between time scales in LEM could be eliminated by reparameterizing. Answering these questions would be enough of a departure from the present analyses that we have decided to postpone them for a future paper.
> >
> > We agree that the perspective of multiscale learning is by no means novel. We cite many papers in ML and neuroscience that have followed variations of this approach. We believe our contributions are (1) expressing the idea in a concise and general form (Eq 2) and proving formal equivalences to other models, and (2) grounding multiscale learning in a Bayesian analysis and then deriving the variational EKF, which significantly extends the basic model.
> >
> > > The authors did not provide code with their submission
> >
> > As we now state in Section 6, we have implemented the variational EKF optimizer in JAX in a form compatible with Optax. This will enable other researchers to use our optimizer in their models with minimal effort. We plan to add a github link to the final de-anonymized version (if accepted), including the code for the optimizer and for the three experiments.

---

### Official Review · Reviewer_x4Pe · 2022-10-25

**Confidence:** 3
**Correctness:** 4
**Technical Novelty And Significance:** 4
**Empirical Novelty And Significance:** 3
**Recommendation:** 5

**Clarity, Quality, Novelty And Reproducibility:**

The writing quality is pretty good throughout -- although it could be even better if it was more streamlined and to the point in the introduction and conclusion. The results are likely to be easily reproducible. I feel that the approach considered is novel, but it still remains largely a mystery how scalable it is based on the submitted draft. I do not feel that the theoretical results showing that this framework generalized past work is that novel or surprising on its own merit. However, if the approach itself was very successful on large scale domains, I would feel that this was a nice part of the discourse.

**Strength And Weaknesses:**

Strengths:
- It is nice that the paper contextualized prior approaches within a common framework.
- I appreciate the synthetic experiments as they help build an understanding of what the model is doing.
- It is nice to show the effectiveness of the variational approach, hinting that the model may showcase scalability.

Weaknesses:
- I did not find the insight about these models being special cases of a common more general framework to be particularly surprising or enlightening. It really only matters, in my view, if it results in a better alternative to the specific models from prior work considered.
- The experiments are extremely limited and the barriers for scaling are not even clearly discussed. The authors end with the comment "the next step will be to investigate how well these methods scale up to large networks, and whether they confer an advantage in real online-learning domains." It feels to me that this is such an obvious step based on the discourse that the current paper comes across as incomplete. If the authors could present experiments showcasing this, I think this paper could have significant impact. However, if this is not possible, I do not believe there will be much interest in the perspective highlighted here. The experiments are so synthetic and toy relative to the typical standards of this conference that I feel that the lack of any experiments on a meaningful scale requires far more justification than provided within the submitted draft.

**Summary Of The Paper:**

The authors make the connection between the framework of multiscale learning and models of fast weights, momentum, and synaptic modelling. The authors then go on to formulate a model of multiscale learning in terms of Bayesian inference over 1/f noise, which has been reported in prior work to match many real-world distributions. The authors test out their model on synthetic online prediction and classification tasks while showing a variational approximation can retain most of the benefits of the full model.

**Summary Of The Review:**

Ultimately, I am on the fence about this paper. I definitely see some good parts about it, but ultimately feel that each strength is somewhat undercut by a corresponding weakness in the current submission. I lean towards rejection at the current time because I feel that the current paper is somewhat incomplete. If large scale experiments were provided validating the proposed method, I would think it could make a very nice contribution to the conference and I would be firmly voting for accept. However, the current experiments are well below the standards of this conference, and not enough justification is provided to make readers understand why this is a logical ending point for the work.

---

> ### Author Response · Authors · 2022-11-18
> **Reply to Reviewer x4Pe**
>
> Thank you for your tentative support and constructive comments. The suggestions in this review were clear and provided helpful guidance on improving the paper. We hope that our responses below and corresponding revisions to the paper convince you that the theoretical parts of the paper are a stronger standalone contribution than may have been apparent in the previous draft. Perhaps more importantly, we hope you’ll agree that the new simulation greatly strengthens the experimental portion of the paper and fills the gap needed for the paper as a whole to make an excellent contribution to the conference proceedings.
>
> > I did not find the insight about these models being special cases of a common more general framework to be particularly surprising or enlightening. It really only matters, in my view, if it results in a better alternative to the specific models from prior work considered.
>
> We’d like to stress that our results in Section 3 are not about defining a more general model that is a superset of existing models. Instead, they show that a simpler model is as expressive as more general ones. That is, one can remove couplings between timescales and still have the same model (up to reparameterization). If this were unsurprising, previous models would not have incorporated this coupling. Likewise, the connection between momentum and fast weights was not previously known, and if it were then these would not be two separate lines of research in the current literature.
>
> We agree that formal connections of this sort are primarily useful only to the extent they offer insights that lead to new models. The second part of the paper (Section 4) does exactly this. The simplicity of the multiscale optimizer makes it easier to recognize that it should be most effective in $1/f$-type environments (via our generative model of $1/f$ noise as a sum of OU processes at different timescales). This motivates our Bayesian analysis and approximate inference methods, which in turn yield a much more sophisticated model (the variational EKF). The variational EKF extends the multiscale optimizer by setting the decay rates based on spectral analysis and adapting the learning rates online based on tracking uncertainty. We have stated these relationships more clearly in the revision, to emphasize how the variational EKF goes beyond existing methods.
>
> Additionally, the present results suggest further developments that are the focus of our ongoing work. We mention these in Section 6: other ways of obtaining a diagonal approximation of the covariance matrix, block-diagonal approximations that retain certain covariance structure, and variations on the spectral analysis that tune the model to the temporal statistics of a given domain.
>
> > The experiments are extremely limited and the barriers for scaling are not even clearly discussed…I feel that the current paper is somewhat incomplete. If large scale experiments were provided validating the proposed method, I would think it could make a very nice contribution to the conference and I would be firmly voting for accept.
>
> This is a fair criticism of the experiments in our previous draft. Although this is primarily a theory paper, and we believe the formal results in Sections 3 and 4 make a strong contribution on their own, we agree that a more realistic test would greatly strengthen the paper. Therefore we have added a third simulation on real data (MNIST with nonstationary label sequences) with a larger CNN as the predictive model. MNIST is certainly modest for contemporary ML problems, but our results nevertheless provide good evidence that our approach scales well in both performance and efficiency. Please see Section 5.3 and the detailed summary in our joint response to all reviewers (item 1).
>
> Concerning scaling, we note in the paper that our approach is not expensive relative to other optimizers such as Adam, which also maintain several variables for each weight. The memory requirements are linear in the size of the network: For each weight, the multiscale optimizer maintains one subweight per time scale, and the variational EKF maintains one subweight and one variance per time scale. Note also that this is a first-order method that requires only the Jacobian of the network, which is calculated similarly to the loss gradient.
>
> > The writing quality is pretty good throughout -- although it could be even better if it was more streamlined and to the point in the introduction and conclusion.
>
> We agree these sections contained some unnecessary material and have shortened them a bit, though with the number of theoretical elements and mathematical tools we bring together we still need a somewhat long introduction to describe our approach clearly and fully. We hope you will find the revised introduction stays on point.

---

### Official Review · Reviewer_5UF5 · 2022-10-25

**Confidence:** 3
**Correctness:** 3
**Technical Novelty And Significance:** 3
**Empirical Novelty And Significance:** 3
**Recommendation:** 6

**Clarity, Quality, Novelty And Reproducibility:**

For me, the paper is hard to follow. The results are however interesting and relevant. The inference techniques used are well-known and standard. However, I think the work is probably interesting for the community for trying to solve a learning problem with time-varying parameters.

**Strength And Weaknesses:**

Overall, I think the paper presents an interesting new algorithm, for a problem that is probably very relevant in many applications. I am not very used to the style of writing, where the authors are probably coming a bit more from the direction of neuroscience. For me, the paper was pretty hard to follow. However, the results look reasonable even though the derivations are derived in a bit nonstandard way. The presented two empirical case studies are fair and are showing promising results. However, I would have hoped for an application where this strategy is tested for a real-world problem, to show its applicability.

A sore point for me is the derivation using the power spectral analysis. I think this paper would have been much easier to understand and interpret would it had been derived using standard Bayesian inference techniques. For example, the definition of the $1/f$ noise process is very strange to me. To illustrate my confusion: equation (13) is a linear stochastic differential equation, for which the time-point-wise marginal densities are distributed multivariate Gaussian, see, e.g., [1]. Since the sum of the components of a multivariate Gaussian random variable in equation (14) is still Gaussian, see, e.g., https://web.ipac.caltech.edu/staff/fmasci/home/astro_refs/SumOfCorrelatedRVs.pdf , it implies that $\xi(t)$ is a Gaussian random variable, where the mean and variance parameter can be computed in closed form. Maybe the authors can explain, why the $1/f$ definition is needed.

Strengths:
- Interesting Setup
- Relation to well-known algorithms is given
- A new learning algorithm, which is easy to implement
- Convincing empirical results for some synthetic problems under the modeling assumption

Weaknesses:
- The derivations are hard to follow
- The $1/f$ noise definition is not clear to me
- No real-world application given in the paper

[1] Särkkä, Simo, and Arno Solin. Applied stochastic differential equations. Vol. 10. Cambridge University Press, 2019.

**Summary Of The Paper:**

The paper proposes a gradient-based learning scheme that deals with changing parameters during learning. For this, the authors give a context of their algorithm to the classical momentum learning strategy and a multi-timescale model from neuroscience. The algorithm is derived using a Bayesian inference framework, where the posterior mean recovers the newly proposed learning method. Here, a continuous-time linear state space model is used to model the time-varying parameters. For the case of a supervised learning problem, the authors consider a nonlinear model for the time-varying parameters.
Bayesian inference is carried out by the authors by a variational approximation of the extended Kalman filter algorithm. The learning algorithm is tested on two synthetic problems, with time-varying parameters. For both linear regression and classification problems, the proposed algorithm leads to a smaller average loss.

**Summary Of The Review:**

The paper presents an interesting topic. The resulting algorithm is easy to implement and is probably useful for learning problems with time-varying parameters. The derivations are not very clear to me and the paper, but the results look reasonable. An application to a real-world learning problem would have been very helpful to show its applicability.

---

> ### Author Response · Authors · 2022-11-18
> **Reply to Reviewer 5UF5**
>
> Thank you for the overall positive assessment and constructive comments. We’ve addressed all four of your suggestions, as detailed below. We believe the paper is significantly clearer and stronger as a result.
>
> > I am not very used to the style of writing, where the authors are probably coming a bit more from the direction of neuroscience. For me, the paper was pretty hard to follow.
>
> Apologies for adopting more of a neuroscience style of organization (e.g., in the introduction). We hope the revisions we’ve made in response to your other comments have made the paper easier to follow.
>
> > The presented two empirical case studies are fair and are showing promising results. However, I would have hoped for an application where this strategy is tested for a real-world problem, to show its applicability.
>
> We completely agree about the importance of applicability. Although this is primarily a theory paper and we believe the strongest contributions are in the analytic results, we have added a third simulation on a real-world problem (MNIST with nonstationary label sequences) with a significantly larger CNN as the predictive model. Please see Section 5.3 and the detailed summary in our joint response to all reviewers (item 1).
>
> > A sore point for me is the derivation using the power spectral analysis. I think this paper would have been much easier to understand and interpret would it had been derived using standard Bayesian inference techniques.
>
> Thank you for pointing this out. We think the problem was that we didn’t adequately separate the generative and inferential parts of the Bayesian model (see item 3 of our joint response to all reviewers). All the machinery based on Ornstein-Uhlenbeck processes and power spectra lies on the generative side. It serves as a construction or representation of $1/f$ noise that supports efficient inference. The inferential side of the Bayesian model is totally standard, using the framework of the Kalman filter. We have reorganized Section 4 accordingly: Section 4.1 (and Appendix B) describes the generative model of $1/f$ noise, Section 4.2 (and Appendices C-E) describes the inferential side and the extended Kalman filter, and Section 4.3 (and Appendix F) describes the variational approximation to the EKF.
>
> In a bit more detail, one of our main innovations is a Bayesian learner that embodies an expectation for $1/f$ noise yet admits tractable inference. We do this by equipping the learner with the generative model in Eqs 13-14, where the variable in question ($\xi$) is a sum of OU processes at different timescales ($z_i$). Under this generative model, $\xi$ is a $1/f$ noise process (this is the import of the spectral analysis) yet exact Bayesian inference is possible via a Kalman filter over the latent vector $\boldsymbol{Z}$.
>
> > Equation (13) is a linear stochastic differential equation, for which the time-point-wise marginal densities are distributed multivariate Gaussian. Since the sum of the components of a multivariate Gaussian random variable in equation (14) is still Gaussian, it implies that ξ(t) is a Gaussian random variable, where the mean and variance parameter can be computed in closed form. Maybe the authors can explain, why the 1/f definition is needed.
>
> We explain this now in Section 4.1. You are correct that the timepoint marginals are all Gaussian, but what matter for our purposes are the correlations between timepoints. These correlations are what determine the form of the nonstationarity. In other words, the OU process in Eq 13 and the $1/f$ process in Eq 14 are both Gaussian processes, but their kernels are qualitatively different. As we write now, the OU process has exponentially decaying (short range) correlations, $\mathbb{E}[z_i(t)z_i(t+s)] \propto e^{-s/\tau_i}$, whereas the $1/f$ process has power-law (long range) correlations, $\mathbb{E}[\xi(t)\xi(t+s)] \propto s^{-1}$ (see text after Eq 13 and after Eq 14).

---

### Author Response · Authors · 2022-11-18
**Response to all reviewers (part 1 of 2)**

Many thanks to all four reviewers for your supportive and constructive comments. To help you efficiently assess the changes to our paper, we have color-coded the revision with red for new text and green for text that has moved.

There were three main themes in the reviews, which sparked most of our revisions. We believe these major changes have significantly strengthened and sharpened our contribution:

1. More realistic testing

2. Separating our modeling methods (for approximate Bayesian inference) from our simulation methods (for generating nonstationary learning tasks)

3. Separating the Bayesian model’s generative assumptions (involving diffusion processes and power spectra) from its inferential mechanisms (extended Kalman filter and variational approximation)

We describe these three points in detail in the remainder of this reply. We have also carefully addressed all other comments, as described in our separate responses to the individual reviews. We believe these changes have further improved the paper, and we appreciate the time and thought that went into the reviews.

### 1. More realistic testing

As the reviewers all appreciate, this is primarily a theory paper, with contributions lying in (1) formal relationships between the multiscale optimizer and previous (superficially different or more complicated) models, (2) the theory of Bayesian inference over $1/f$ noise, and (3) the variational EKF that extends the multiscale optimizer by approximate Bayesian tracking of uncertainty to adapt learning rates.

From this perspective, the experiments are mainly intended to illustrate the theory. Nevertheless, we agree with the reviewers that the two experiments included in our first submission were simpler than would be desired. There are important questions about how our methods scale up, including (1) how the approximations in the EKF and diagonal variational inference fare with a more complex nonlinear model such as a larger, multilayer NN; (2) whether the algorithm remains efficient in these more complex settings; and (3) how robust it is to deviations between the true dynamics in the environment and the assumed generative model for $1/f$ noise, and between the true data-generating process and the predictive model $y=h(\boldsymbol{x},\boldsymbol{\theta})$.

To address all of these questions, we have conducted a slightly larger-scale experiment with a CNN and MNIST. As described in Section 5.3, we devised a nonstationary classification task whereby the item on each time step was sampled according to class probabilities that varied over time, specifically using logits that followed $1/f$ noise. This setup reflects the idea that any realistic data stream for real-time classification will feature autocorrelations, and these autocorrelations will typically span multiple time scales.

Regarding question 1 above, we find the variational EKF performs very well in this setting, reaching 99% accuracy on new items after only ¼ of an epoch (not reported). This should be interpreted as test accuracy since each item is being seen for the first time. As we report in the paper (see Figure 5), the variational EKF outperforms SGD (with optimized learning rate and momentum), and most importantly it can better exploit the temporal structure, as seen by comparisons between $1/f$ and iid sequences.

Regarding question 2, we find compute time scales linearly with network size and with the number of time scales used ($n$), as expected (not reported). Surprisingly, our custom Optax-style optimizer code for the variational EKF, with 8 time scales, is slightly faster than Optax’s off-the-shelf SGD optimizer. The comparison is not strictly fair, and certainly SGD would be as fast or faster than our optimizer if the code were matched line for line. Nevertheless, these observations suggest that compute time is not a serious impediment to scaling.

Regarding question 3, the architecture of the CNN, while expressively flexible, has nothing to do with how the MNIST images were actually produced (i.e., by humans). This is in contrast to the first two experiments, where the predictive model was matched or closely related to the data generation. Additionally, the sampling procedure used to create the label sequence, described in Appendix G, was different from the generative model of $1/f$ noise assumed by our Bayesian analysis (the sum of OU processes in Eq 16 and Appendix B). These observations suggest our approach may scale well to larger ML problems, where the data-generating process and temporal dynamics are typically unknown.

---

> ### Author Response · Authors · 2022-11-18
> **Response to all reviewers (part 2 of 2)**
>
> ### 2. Separating modeling methods from simulation methods
>
> As is standard with Bayesian models, our model posits a latent casual structure for explaining observed data. In the present case, this generative model is given by the sum of diffusion (Ornstein-Uhlenbeck, OU) processes described in Section 4.1. Importantly, the latent variables ($z_i$) exist only within the Bayesian model and need not correspond to anything real in the environment. Again, this is standard for real applications of Bayesian modeling, where one typically does not know the true data-generating process but hypothesizes some generative model that one hopes is accurate and expressive enough to support successful inference.
>
> In the revision, we have been careful to separate the Bayesian generative model from the true data-generating process. In referring to the decomposition of $\boldsymbol{\theta}$ as a sum of OU processes, we emphasize that this is an assumption inside the Bayesian model (e.g., after Eq 15: “the generative side of our Bayesian model posits latent variables $\boldsymbol{z}_i$...”). In referring to our procedures for data-generation in the experiments, we use different notation, for example writing $\boldsymbol{\beta}$ for the true regression coefficients in Section 5.1, to distinguish them from the model’s posited variables $\boldsymbol{\theta}$ and $\boldsymbol{z}$ and its mean estimates $\boldsymbol{w}$ and $\boldsymbol{\omega}$.
>
> ### 3. Separating generative assumptions from inferential mechanisms
>
> Our revision also better separates the generative model of $1/f$ noise described above from the inferential side of the Bayesian model. The generative model is founded on tools from the theory of stochastic processes and spectral analysis (OU processes, Fourier transforms, and power spectra). Under our revised organization, this material is all contained in Section 4.1 and Appendix B. None of this is related to Bayesian inference per se. The inferential side of the model is the Kalman filter, which we then extend to the EKF and variational approximation (Sections 4.2-4.3 and Appendices C-F).
>
> One of our key innovations in developing a model of Bayesian inference over $1/f$ noise lies in linking the generative and inferential sides of the model. The bridge is the expanded state representation, $\boldsymbol{Z}=(\boldsymbol{z}_1,\dots,\boldsymbol{z}_n)$. On the generative side, it describes a family of stochastic processes that, when summed, yield an aggregate $1/f$ process. On the inferential side, $\boldsymbol{Z}$ is a linear dynamic system that supports efficient Bayesian inference via the Kalman filter. We hope this is all clearer under the new organization of Section 4.

---

### Decision · Program_Chairs · 2023-01-20

**Decision:**

Accept: poster

**Justification For Why Not Higher Score:**

The overall ratings by reviewers were not that high.

**Justification For Why Not Lower Score:**

A majority of reviewers voted for acceptance.

**Metareview: Summary, Strengths And Weaknesses:**


Summary:

The paper proposes a gradient-based learning scheme that deals with changing parameters during learning. For this, the authors give a context of their algorithm to the classical momentum learning strategy and a multi-timescale model from neuroscience. The algorithm is derived using a Bayesian inference framework, where the posterior mean recovers the newly proposed learning method. Here, a continuous-time linear state space model is used to model the time-varying parameters. For the case of a supervised learning problem, the authors consider a nonlinear model for the time-varying parameters. Bayesian inference is carried out by the authors by a variational approximation of the extended Kalman filter algorithm. The learning algorithm is tested on two synthetic problems, with time-varying parameters. For both linear regression and classification problems, the proposed algorithm leads to a smaller average loss.

Strengths:

- Relation to well-known algorithms is given
- A new learning algorithm, which is easy to implement
- Convincing empirical results for some synthetic problems under the modeling assumption
- It is nice that the paper contextualized prior approaches within a common framework.
- I appreciate the synthetic experiments as they help build an understanding of what the model is doing.
- It is nice to show the effectiveness of the variational approach, hinting that the model may showcase scalability.
- The writing quality is pretty good throughout
- The paper tackles an important problem of learning in non-i.i.d settings, in the specific case where multiple learning processes operate at different timescales.
- The authors present a unifying framework of multiscale optimizers for learning in such settings. The framework encompasses well studied approaches of momentum learning and the Benna-Fusi model, providing novel insights for momentum learning.
- The Bayesian formulation provides a simple and straightforward approach for implementing the multiscale optimizer.
- The synthetic simulations provide some evidence on the applicability of the multiscale optimizer in practical settings. (I discuss this a bit more in the weakness)

Weaknesses:

- The derivations are hard to follow
- The 1/f noise definition is not clear
- No real-world application given in the paper
- The experiments are extremely limited.
- it still remains largely a mystery how scalable it is based on the submitted draft.
- The paper studies a particular setting where the multi-scale weights are additive. This however seems like a somewhat limiting assumption to me. The authors do not discuss this choice in the paper and it is not clear whether this assumption holds true in realistic settings.
- While the simulations provide a sanity check for the approach, they leave quite a bit to be desired. The assumption of weights of the weights of the different timescales being additive might not hold in any realistic setting. Experiments on more realistic settings would be helpful to address this concern.

Recommendation:

A majority of reviewers vote for acceptance. The reviewer voting for rejection asked for large scale experiments. These were added in the rebuttal. I, therefore, decide to accept the paper. I encourage the authors to use the feedback provided to improve the paper for its camera ready version.


**Note From Pc:**

if the above contains the word "oral" or "spotlight" please see: "oral" presentation means -> notable-top-5% and "spotlight" means -> notable-top-25%. As stated in our emails, we are disassociating presentation type from AC recommendations